# Gas6 ameliorates intestinal mucosal immunosenescence to prevent the translocation of a gut pathobiont, *Klebsiella pneumoniae*, to the liver

Hitoshi Tsugawa[ID][1]*, Takuto Ohki[2], Shogo Tsubaki[1], Rika Tanaka[3], Juntaro Matsuzaki[4], Hidekazu Suzuki[5], Katsuto Hozumi[3]

1 Transkingdom Signaling Research Unit, Division of Host Defense Mechanism, Tokai University School of Medicine, Isehara, Japan, 2 Department of Hand Surgery, Nagoya University Graduate School of Medicine, Nagoya, Japan, 3 Department of Immunology, Division of Host Defense Mechanism, Tokai University School of Medicine, Isehara, Japan, 4 Division of Pharmacotherapeutics, Keio University Faculty of Pharmacy, Tokyo, Japan, 5 Division of Gastroenterology and Hepatology, Department of Internal Medicine, Tokai University School of Medicine, Isehara, Japan

* tsugawa.hitoshi.r@tokai.ac.jp

**Data Availability Statement:** All relevant data are within the manuscript and its Supporting Information files.

## Abstract

Immunosenescence refers to the development of weakened and/or dysfunctional immune responses associated with aging. Several commensal bacteria can be pathogenic in immunosuppressed individuals. Although *Klebsiella pneumoniae* is a commensal bacterium that colonizes human mucosal surfaces, the gastrointestinal tract, and the oropharynx, it can cause serious infectious diseases, such as pneumonia, urinary tract infections, and liver abscesses, primarily in elderly patients. However, the reason why *K. pneumoniae* is a more prevalent cause of infection in the elderly population remains unclear. This study aimed to determine how the host's intestinal immune response to *K. pneumoniae* varies with age. To this end, the study analyzed an *in vivo K. pneumoniae* infection model using aged mice, as well as an *in vitro K. pneumoniae* infection model using a Transwell insert co-culture system comprising epithelial cells and macrophages. In this study, we demonstrate that growth arrest-specific 6 (Gas6), released by intestinal macrophages that recognize *K. pneumoniae*, inhibits bacterial translocation from the gastrointestinal tract by enhancing tight-junction barriers in the intestinal epithelium. However, in aging mice, Gas6 was hardly secreted under *K. pneumoniae* infection due to decreasing intestinal mucosal macrophages; therefore, *K. pneumoniae* can easily invade the intestinal epithelium and subsequently translocate to the liver. Moreover, the administration of Gas6 recombinant protein to elderly mice prevented the translocation of *K. pneumoniae* from the gastrointestinal tract and significantly prolonged their survival. From these findings, we conclude that the age-related decrease in Gas6 secretion in the intestinal mucosa is the reason why *K. pneumoniae* can be pathogenic in the elderly, thereby indicating that Gas6 could be effective in protecting the elderly against infectious diseases caused by gut pathogens.

**Funding:** This study was supported by the Kao Research Council for the Study of Healthcare Science (C93102) (to H.T.). The infection models in this work were partly supported by a 2021 Tokai University School of Medicine Research Aid grant (210049) (to H.T.), Tokai University Research Organization Grant (21NS331825) (to H.T.), Ohyama Health Foundation Inc. (113) (to H.T.), and the Japan Agency for Medical Research and Development (21ck0106701h0001) (to J.M.). The funders had no role in study design, data collection and analysis, decision to publish, or preparation of the manuscript.

**Competing interests:** The authors have declared that no competing interests exist.

## Author summary

Aging causes a weakened/dysfunctional human immune system, reducing the ability to combat pathogens. Understanding the molecular mechanisms underlying age-related immunosenescence is critical for the development of preventive therapies against bacterial infectious diseases in elderly individuals. *Klebsiella pneumoniae* is a representative "pathobiont" that causes serious systemic infections such as pneumonia, urinary tract infections, and liver abscesses, mainly in the elderly. However, it remains unclear how *K. pneumoniae* is a more prevalent cause of infection in the elderly population. Here, we show that growth arrest-specific 6 (Gas6), released by intestinal macrophages that recognize *K. pneumoniae*, inhibits bacterial invasion into the intestinal epithelium and subsequent translocation to the liver. However, in elderly mice, Gas6 is hardly secreted due to decreased intestinal mucosal macrophages; therefore, *K. pneumoniae* can easily translocate to the liver from the gastrointestinal tract. We concluded that the reason *K. pneumoniae* can be pathogenic to the elderly is the age-related decrease in Gas6 secretion in the intestinal mucosa. Moreover, we revealed that the administration of Gas6 to elderly mice significantly prevented systemic translocation of orally infected *K. pneumoniae*. Our findings provide new insights into the prevention of infectious diseases in the elderly.

## Introduction

Weakened and/or dysfunctional immune responses caused by aging, results in a reduced ability to combat pathogens [1]. Certain infections are more prevalent in the elderly than in younger adults, and there is no doubt that compromised immune function in the elderly is strongly related to a variety of infectious diseases [2]. In particular, the community-acquired pneumonia and urinary tract infections are 3- and 20-fold more common in elderly, respectively [2]. Additionally, the microorganisms that cause infectious disease in the elderly are becoming more diverse; some commensal bacteria that exert pathogenicity in the immunosenescent patients, termed "pathobionts," can become a fatal clinical concern [3].

*Klebsiella pneumoniae* is an encapsulated gram-negative bacterium found in soil, water, and on the surfaces of medical devices [4]. *K. pneumoniae* is a commensal bacterium that colonizes human mucosal surfaces, the gastrointestinal tract, and the oropharynx. It is a gut colonizer with pathogenic potential that causes serious infectious diseases such as pneumonia, urinary tract infections, and liver abscesses in immunocompromised patients, including the elderly [4,5]. In the 1980s, highly virulent strains of *K. pneumoniae* that caused liver infections in healthy individuals were isolated [6,7]. Additionally, *K. pneumoniae* has become increasingly resistant to antibiotics. In fact, infections caused by carbapenem-resistant *K. pneumoniae* have resulted in a significant increase in morbidity and mortality [8]. Therefore, it is crucial preventing infectious diseases caused by *K. pneumoniae* than developing novel antibiotics. Although the pathogenicity of *K. pneumoniae* needs to be understood in detail to design novel strategies to prevent its infection, the detailed mechanisms of host-*K. pneumoniae* interactions associated with the development of these diseases remain unclear. It is unclear why *K. pneumoniae* is a more prevalent cause of infection in the elderly. The aim of this study was to determine how the host's intestinal immune response to *K. pneumoniae* changes with age. In this context, we analyzed an *in vivo K. pneumoniae* infection model using elderly mice and an *in vitro K. pneumoniae* infection model using a Transwell insert co-culture system based on epithelial cells and macrophages. The results showed that growth arrest-specific 6 (Gas6), released

by macrophages that recognize *K. pneumoniae*, enhances the Gas6/Axl axis in the intestinal epithelium. Gas6 is a secreted protein and ligand of the Axl tyrosine kinase receptor. The binding of Gas6 to Axl mediates several biological signals related to cell proliferation, survival, and migration [9]. We showed that Gas6 release in the intestinal mucosa decreases with age, allowing *K. pneumoniae* to readily invade the intestinal epithelium and cause systemic infections. Our findings have the potential to facilitate the development of preventative measures for infectious diseases caused by *K. pneumoniae* in the elderly.

## Results

### *K. pneumoniae* oral infection in elderly mice easily invade the gastrointestinal submucosa and translocate to the liver

To examine age-related differences in susceptibility to *K. pneumoniae*, 15- or 57-week-old mice were given antibiotic-laced drinking water and orally infected with *K. pneumoniae* (**Fig 1A**). As shown in S1A and S1B Fig, no difference of the cecal mucosal architecture and thickness detected between 15-week-, and 57-week-old mice before initiation of the antibiotic treatment. To distinguish between endogenous (a gut commensal) *K. pneumoniae* and exogenous (orally introduced) *K. pneumoniae*, we electroporated *K. pneumoniae* ATCC43816 with the pmCherry plasmid (**S1C Fig**) and then infected the mice. Infection with *K. pneumoniae* ATCC43816 pmCherry severely reduced the survival of 57-week-old mice compared to 15-week-old mice (**Fig 1B**). At 2 days post-infection, we observed desquamation of the epithelial cells in the cecal mucosal layer (**S1D, black dotted line, and S1E Fig**) and edema of the cecal submucosa in 57-week-old mice, but not in 15-week-old mice (**S1D, red dotted line, and S1E Fig**). Moreover, we observed liver damage in 57-week-old mice infected with *K. pneumoniae* ATCC43816 pmCherry but not in 15-week-old mice (**S1F and S1G Fig**). In an experiment to determine the bacterial number in infected mice, orally infected *K. pneumoniae* ATCC43816 pmCherry was selectively measured by culturing on ampicillin-containing LB plates. The number of bacteria in the cecal mucosa and liver of 57-week-old mice 2 days post-infection was significantly higher than that in 15-week-old mice (**Fig 1C**). Immunostaining with an anti-mCherry antibody detected orally infected *K. pneumoniae* on the surface of cecal epithelial cells in 15-week-old mice; however, these signals in 57-week-old mice were detected within the cecal submucosa, indicating that orally infected *K. pneumoniae* were only able to invade the cecal mucosal layer of elderly mice (**Fig 1D and 1E**). Moreover, signals generated by *K. pneumoniae* were much stronger in the livers of 57-week-old mice than in those of 15-week-old mice, indicating that orally infected *K. pneumoniae* in 57-week-old mice translocated easily to the liver (**Fig 1F and 1G**).

### Macrophages upregulate the expression of tight junction proteins between epithelial cells under *K. pneumoniae* infection and prevent bacterial invasion

To determine why *K. pneumoniae* invasion into the intestinal mucosa of young adult mice was inhibited, we constructed an *in vitro K. pneumoniae* infection model using a Transwell insert co-culture system based on Caco-2 epithelial cells and RAW264.7 macrophages or murine bone marrow-derived macrophages (BMDMs) (**Fig 2A**). Caco-2 cells co-cultured with RAW264.7 macrophages or BMDMs in the Transwell insert co-culture system were infected with *K. pneumoniae* for 1.5 h. Next, the Caco-2 cells were incubated for 6 h in Dulbecco's modified Eagle's medium (DMEM) containing 100 μg/mL gentamycin to kill extracellular bacteria (**Fig 2A**). Hematoxylin and eosin (H&E) staining revealed that Caco-2 cells were not affected

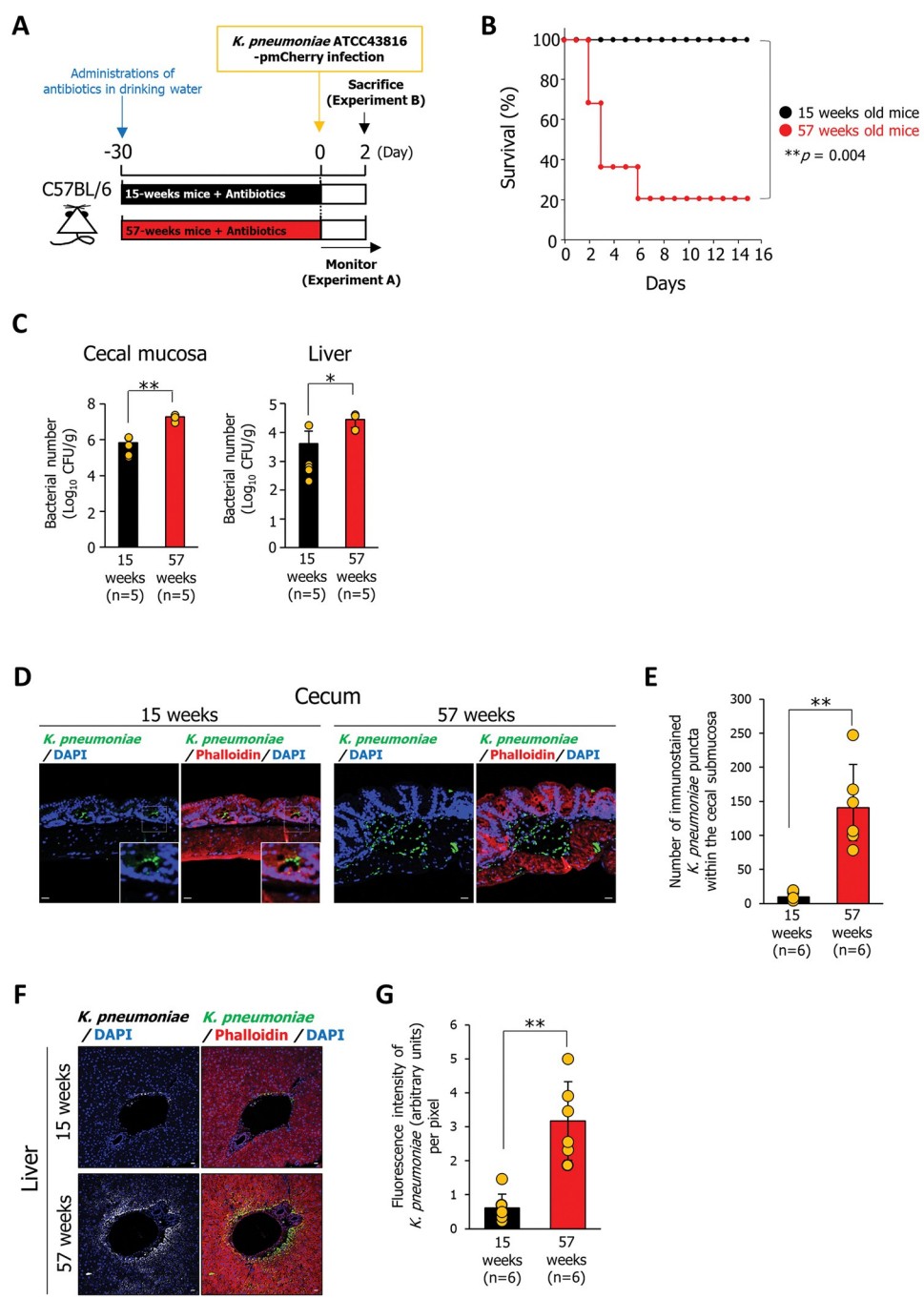

**Fig 1. Oral infection with *K. pneumoniae* is highly pathogenic in elderly mice but not in young adult mice. (A)** Treatment scheme used to analyze age-related differences in susceptibility to *K. pneumoniae*. Mice aged 15- or 57-week were administered antibiotics for 4 weeks prior to oral inoculation of *K. pneumoniae* ATCC43816 pmCherry ($5 \times 10^7$ bacteria). Survival was monitored daily (Experiment A). At 2 days post-infection, the mice were sacrificed, and the cecum and liver were harvested (Experiment B). **(B)** Effect of aging on the survival of mice infected with *K. pneumoniae* ATCC43816 pmCherry. Each dot was represented from an individual mouse (n = 6 per group). *p*-values were determined using the log-rank test. **(C)** Bacterial counts in the cecum and liver were determined 2 days after infection. Cecum and liver tissues were homogenized in PBS, and the homogenates were plated onto LB agar containing 400 μg/mL ampicillin. The number of CFU was determined. Each dot represents the value from an individual mouse (n = 5 per group). Data are presented as the mean ± SD. *$p < 0.05$, **$p < 0.01$. *p* values were calculated by Student's *t* test. **(D and F)** Sections of cecal mucosa (D) or liver (F) were from mice aged 15 or 57 weeks at 2 days after infection with *K. pneumoniae* ATCC43816 pmCherry and immunostained with an anti-mCherry antibody and rhodamine phalloidin staining. Each image has six independent replicates. Scale bar = 20 μm. **(E)**

Number of immunostained puncta of *K. pneumoniae* within the cecal submucosa per same area were counted by the ImageJ analysis software. Each dot represents the value from an individual mouse (n = 6 per group). Data are presented as the mean ± SD. \*\**p* < 0.01. *p* values were calculated by the Student's *t* test. **(G)** Fluorescence intensity of immunostained *K. pneumoniae* per pixel were measured by the ImageJ analysis software. Each dot represents the value from an individual mouse (n = 6 per group). Data are presented as the mean ± SD. \*\**p* < 0.01. *p* values were calculated by the Student's *t* test.

by *K. pneumoniae* infection under the co-culture conditions with RAW264.7 cells employed; however, in the absence of RAW264.7 macrophages, *K. pneumoniae* ruptured Caco-2 cells (**S2A and S2B Fig**). Furthermore, in the absence of RAW264.7 or BMDMs, the number of bacteria invading Caco-2 cells was significantly higher than that in the presence of RAW264.7 or BMDMs, suggesting that macrophages inhibit bacterial invasion into epithelial cells (**Fig 2B and 2C**). Moreover, immunostaining of *K. pneumoniae* reaching the basolateral side of Caco-2 cells was detected in the absence of RAW264.7 macrophages (**Fig 2D**). Expression of tight junction proteins ZO-1 and occludin of Caco-2 cells infected with *K. pneumoniae* in the absence of RAW264.7 macrophages was significantly lower than that in the presence of RAW264.7 macrophages (**Fig 2E, 2F, 2G and 2H**). The ZO-1 and occludin expression in *K. pneumoniae*-uninfected Caco-2 cells was not decreased even in the absence of RAW264.7 macrophages (**S3A, S3B, S3C and S3D Fig**).

## Gas6 and Axl are released by macrophages that recognize *K. pneumoniae* infection, and the Gas6 is co-localized with the Axl tyrosine kinase receptor on the epithelial cells

To explore how RAW264.7 macrophages regulate the expression of tight junction proteins involved in the repression of bacterial invasion, we focused on cytokines released during *K. pneumoniae* infection. After exposure to *K. pneumoniae* for 1.5 h, Caco-2 cells grown in a Transwell co-culture system with RAW264.7 macrophages were incubated for 6 h with DMEM containing 100 μg/mL gentamycin. The culture medium was harvested for cytokine array analysis. Cytokine array analysis revealed that the *K. pneumoniae* infection significantly increased the release of the receptor tyrosine kinase Axl (**Fig 3A**). To confirm this result, the culture supernatant or the bacterial lysate from *K. pneumoniae* was added to RAW264.7 or Caco-2 cells, and the culture medium was collected for Axl measurement using an ELISA. The secretion of Axl by RAW264.7 macrophages after exposure to *K. pneumoniae* culture supernatant or bacterial lysate increased significantly in a dose-dependent manner (**Fig 3B**). Axl secretion by Caco-2 cells was not observed (**Fig 3B**). Gas6 is an Axl ligand. Gas6/Axl signaling, which is triggered by the binding of Gas6 to Axl, is involved in cell proliferation, survival, and migration [9]. Therefore, we investigated whether *K. pneumoniae* enhanced Gas6 secretion by RAW264.7 macrophages. Gas6 secretion by RAW264.7 macrophages increased significantly in a dose-dependent manner upon exposure to *K. pneumoniae* culture supernatant or bacterial lysate; however, Caco-2 cells did not exhibit an increase in Gas6 secretion (**Fig 3C**). Interestingly, heat-treated (95˚C, 5 min) bacterial supernatants or lysates also induced Gas6 and Axl secretion by RAW264.7, indicating that the Axl- and Gas6-inducing factors produced by *K. pneumoniae* are heat-resistant (**Fig 3B and 3C**). Moreover, we observed co-localization of Gas6 and Axl in Caco-2 cells infected with *K. pneumoniae* in the presence of RAW264.7 macrophages (**Fig 3D and 3E**). Co-localization of Gas6 and Axl in Caco-2 cells was not detected in the absence of *K. pneumoniae* or in the presence of *K. pneumoniae* without RAW264.7 macrophages (**Fig 3D and 3E**).

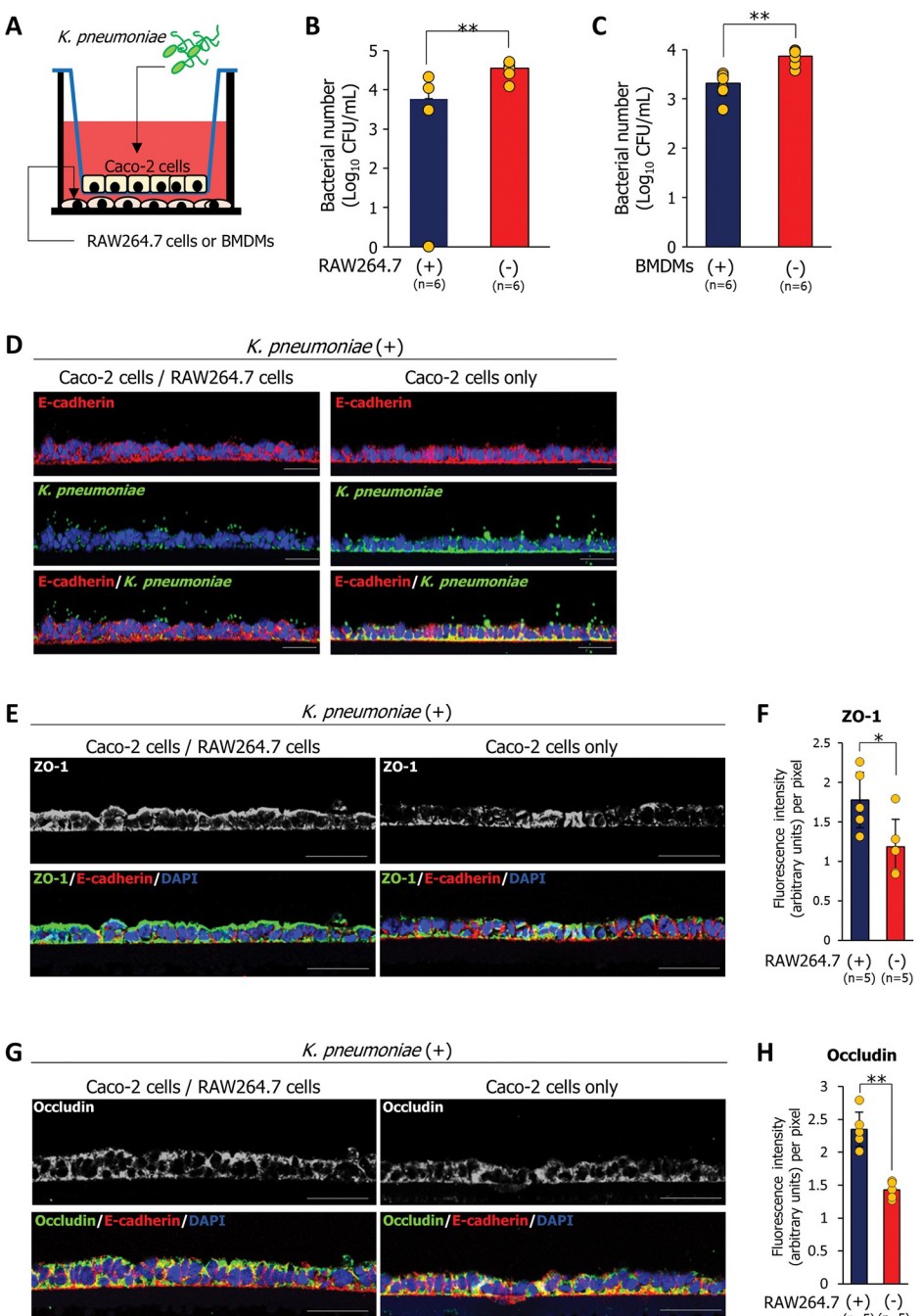

**Fig 2. Macrophages inhibit the invasion of intestinal epithelial cells by _K. pneumoniae in vitro_. (A)** An _in vitro K. pneumoniae_ infection model using a Transwell insert co-culture system consisting of Caco-2 cells grown on the insert and RAW264.7 macrophages or BMDMs grown in the cell culture well. **(B and C)** Bacterial counts in Caco-2 cells in the presence or absence of RAW264.7 macrophages (B) or BMDMs (C). Caco2 cells were lysed with PBS/1% Triton X-100, and the lysate was plated on LB agar. The number of CFU was counted after 24 h incubation. Each dot represents six independent replicates (n = 6 per group). Data are presented as the mean ± SD. **$p < 0.01$. _p_ values were calculated by the Student's _t_ test. **(D)** Caco-2 cells grown on the insert in the presence or absence of RAW264.7 macrophages were immunostained with an anti-E-cadherin antibody and an anti-_Klebsiella pneumoniae_ antibody. Each image has five independent replicates. Scale bar = 50 μm. **(E and G)** Caco-2 cells grown on the insert in the presence or absence of RAW264.7 macrophages were immunostained with an anti-E-cadherin antibody, and an anti-ZO-1 antibody (E) or an anti-occludin antibody (G). Each image has five independent replicates. Scale bar = 50 μm. **(F and H)** Fluorescence intensity per pixel was measured by ImageJ analysis software. Each dot represents five independent replicates (n = 5 per group). Data are presented as the mean ± SD. **$p < 0.01$. _p_ values were calculated by the Student's _t_ test.

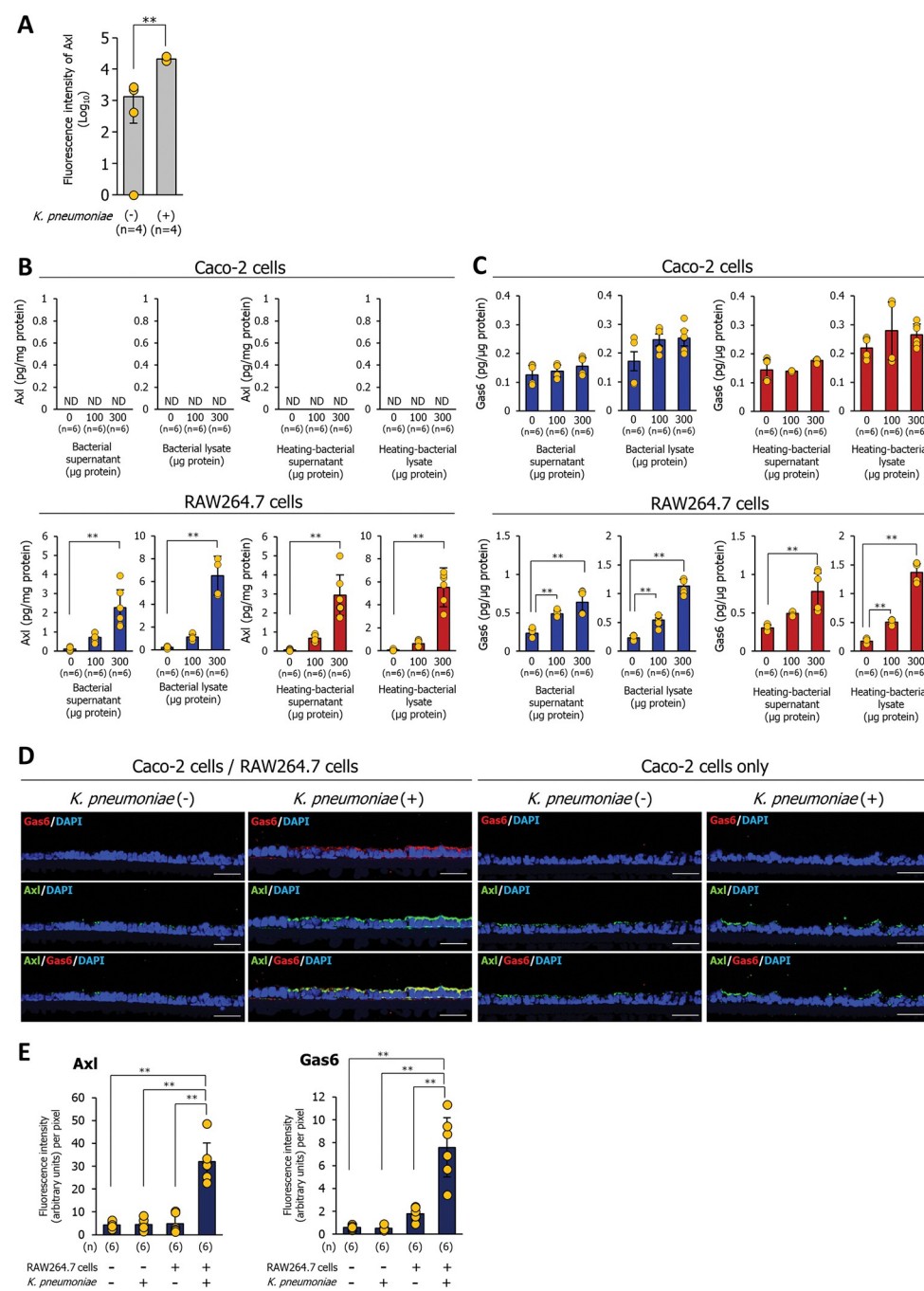

**Fig 3. Gas6 released by macrophages that recognized *K. pneumoniae* infection is co-localized with Axl tyrosine kinase receptor on the epithelial cells. (A)** Cytokine array analysis of the *K. pneumoniae* infection model based on the Transwell insert co-culture system consisting of Caco-2 cells and RAW264.7 macrophages. Quantification of Axl signals using a laser scanner. Each dot represents four independent replicates (n = 4 per group). Data are presented as the mean ± SD. **$p < 0.01$. $p$ values were calculated by the Student's $t$ test. **(B and C)** Secretion of Axl (B) and Gas6 (C) by Caco-2 cells or RAW264.7 macrophages. Culture supernatants or lysates of *K. pneumoniae* were added to Caco-2 cells or RAW264.7 macrophages for 12 h. The culture media were then collected for ELISA analysis of Axl or Gas6 levels. Each dot represents six independent replicates (n = 6 per group). Data are presented as the mean ± SD. **$p < 0.01$. $p$ values were calculated by one way analysis of variance. **(D)** *K. pneumoniae*-infected Caco-2 cells grown on a Transwell insert in the presence or absence of RAW264.7 macrophages were immunostained with anti-Gas6 and anti-Axl antibodies. Each image has six independent replicates. Scale bar = 50 μm. **(E)** Fluorescence intensity per pixel of anti-Axl antibody (left panel) or an anti-Gas6 antibody (right panel) were measured by the ImageJ analysis software. Each dot represents six independent replicates (n = 6 per group). Data are presented as the mean ± SD. **$p < 0.01$. $p$ values were calculated by one way analysis of variance.

## Gas6 released by macrophages, prevents bacterial invasion by upregulating the expression of tight junction proteins between epithelial cells

Based on these results, we hypothesized that the released Gas6 prevents bacterial invasion by increasing the expression of ZO-1 and occludin in Caco-2 cells via binding to Axl. To test this, Caco-2 cells were treated with an Axl inhibitor (R428; 20 nM) or 1 μg of anti-Gas6 antibody for 3 h prior to infection with *K. pneumoniae*. Upon *K. pneumoniae* infection, the expression of Axl, Gas6, ZO-1, and occludin in Caco-2 cells was significantly increased by the presence of RAW264.7 macrophages (**Fig 4A; lanes 1 and 2, and 4B**) or BMDMs (**S4A; lanes 1 and 2, and S4B Fig**). Axl inhibitor (R428) or anti-Gas6 antibody significantly attenuated the increase in the expression of Axl, ZO-1, and occludin of Caco-2 cells in the presence of RAW264.7 macrophages (**Fig 4A; lanes 4 and 6, and 4B**) or the presence of BMDMs (**S4A; lanes 4 and 6, and S4B Fig**). Immunostaining analysis of Caco-2 cells infected with *K. pneumoniae* in the presence of RAW264.7 macrophages revealed a significant decrease in the expression of ZO-1 and occludin by the addition of the Axl inhibitor or anti-Gas6 antibody (**Fig 4C and 4D**). By comparison, in uninfected Caco-2 cells, the expression of ZO-1 and occludin was not affected by the addition of the Axl inhibitor or anti-Gas6 antibody (**S5A and S5B Fig**). Next, we examined the effect of an Axl inhibitor or an anti-Gas6 antibody on the invasion of *K. pneumoniae* into Caco-2 cells. The presence of RAW264.7 significantly inhibited the invasion of *K. pneumoniae* within Caco-2 cells, and this effect was abolished upon treatment with an Axl inhibitor or anti-Gas6 antibody (**Fig 4E and 4F**). Additionally, the decreased invading bacterial number in the presence of BMDMs was eliminated by treatment with an Axl inhibitor or anti-Gas6 antibody (**Fig 4G**). These results suggest that macrophage-released Gas6 enhances the tight junction barrier between epithelial cells to prevent *K. pneumoniae* invasion.

## Expressions of Gas6 and Axl in the cecal mucosa under *K. pneumoniae* infection decreased with aging

We then investigated whether the expression levels of Gas6 and Axl in *K. pneumoniae* infection differ with age. Immunostaining of Gas6 and Axl was stronger in the cecal mucosa of 15-week-old mice infected with *K. pneumoniae* ATCC43816 pmCherry than that in 57-week-old mice, and Gas6 and Axl co-localized in cecal epithelial cells (**Figs 5A and S6A**). Interestingly, co-localization was detected at both the apical and basolateral sides of the epithelial cells (**Fig 5A**). In contrast, Gas6 was barely detectable in the liver epithelial cells of 15-week-old or 57-week-old mice infected with *K. pneumoniae* ATCC43816 pmCherry (**Figs 5B and S6B**). Western blotting confirmed a significant decrease in the expression of Axl, Gas6, ZO-1, and occludin in the cecal mucosa of 57-week-old mice compared to that in 15-week-old mice infected with *K. pneumoniae* ATCC43816 pmCherry (**Fig 5C and 5D**). However, in the liver, the expression of these molecules did not decrease with age (**Fig 5C and 5D**). In addition, we found a significant linear correlation between Gas6 and Axl expression in the cecal mucosa of both 15-week-old and 57-week-old mice (15-week-old mice; $r = 0.957$, 57-week-old mice; $p = 0.0106$, and $r = 0.996$, $p = 0.0003$), but not in the liver (**Fig 5E**). Our findings suggest that age-related decrease in the expression of Gas6 and Axl in the intestinal mucosa increases susceptibility to *K. pneumoniae* oral infection. The number of macrophage precursors and macrophages in the bone marrow of the elderly are reported to be significantly reduced [10,11]. We then examined how the proportion of intestinal mucosal macrophages changes with age. The proportion of F4/80+ macrophages in the intestinal mucosa of aged mice (56-week-old) was 8.87%, whereas it was 29.5% in young mice (14-week-old) (**Fig 5F**). These results suggest that an age-related decrease in intestinal mucosal macrophages may be one of the reasons for reduced Gas6 expression in elderly mice infected with *K. pneumoniae*.

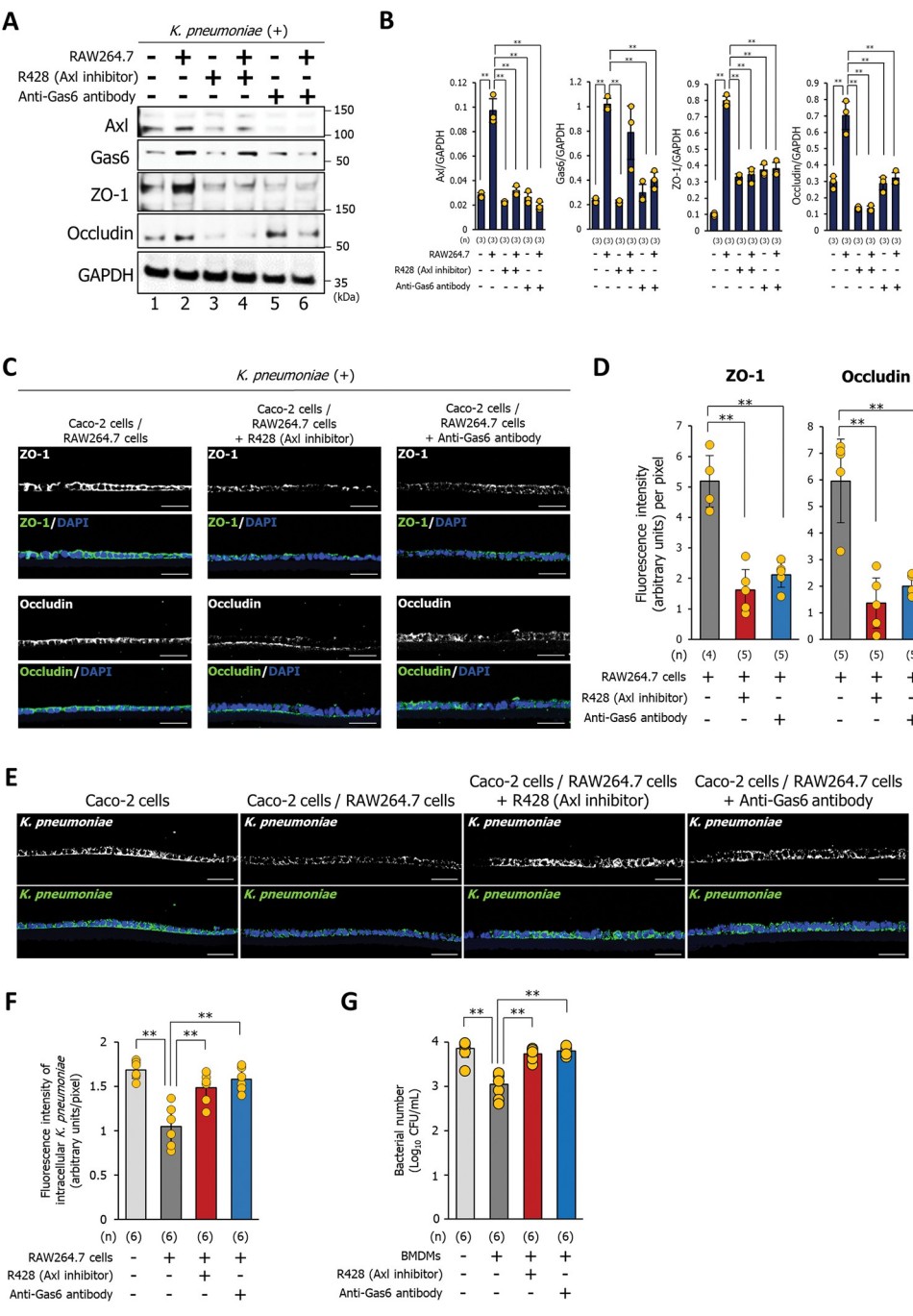

**Fig 4. Gas6/Axl signaling in Caco-2 cells inhibits _K. pneumoniae_ invasion by enhancing the expression of tight junction proteins. (A)** Western blotting was performed to detect the expression of Axl, Gas6, ZO-1, and occludin in Caco-2 cells infected with _K. pneumoniae_ in the presence of an Axl inhibitor (R428) or an anti-Gas6 antibody. Prior to _K. pneumoniae_ infection, cells were treated for 3 h with Axl inhibitor (R428; 20 nM) or 1 μg of anti-Gas6 antibody. Each western blotting image represents three independent replicates (n = 3). **(B)** Western blotting signal intensity was analyzed by ImageJ software. Each dot represents three independent replicates (n = 3). Data are presented as the mean ± SD. **$p < 0.01$. $p$ values were calculated by one way analysis of variance. **(C)** Axl inhibitor (R428; 20 nM) or 1 μg of anti-Gas6 antibody were added to Caco-2 cells grown on the insert in the presence of RAW264.7 macrophages for 3 h prior to _K. pneumoniae_ infection. Next, Caco-2 cells were immunostained with an anti-ZO-1 antibody and an anti-occludin antibody. Each image has four (group of Caco-2 cells/RAW264.7 cells stained with an anti-ZO-1 antibody) or five independent replicates. Scale bar = 50 μm. **(D)** Fluorescence intensity per pixel of anti-ZO-1 antibody (left panel) or an anti-occludin antibody (right panel) were measured by the ImageJ analysis software. Each dot represents four or five independent replicates (n = 4 or 5 per group). Data are presented as the mean ± SD. **$p < 0.01$.

*p* values were calculated by one way analysis of variance. (**E**) Axl inhibitor (R428; 20 nM) or 1 µg of anti-Gas6 antibody were added to Caco-2 cells grown on the insert in the presence or absence of RAW264.7 macrophages for 3 h prior to *K. pneumoniae* infection. Next, Caco-2 cells were immunostained with an anti-*Klebsiella pneumoniae* antibody. Each image has six independent replicates. Scale bar = 50 µm. (**F**) Fluorescence intensity per pixel of *K. pneumoniae* were measured by the ImageJ analysis software. Each dot represents six independent replicates (n = 6 per group). Data are presented as the mean ± SD. **$p < 0.01$. *p* values were calculated by one way analysis of variance. (**G**) Bacterial counts within Caco-2 cells treated with an Axl inhibitor (R428) or an anti-Gas6 antibody in the presence of BMDMs. Prior to *K. pneumoniae* infection, Caco-2 cells were treated for 3 h with Axl inhibitor (R428; 20 nM) or 1 µg of anti-Gas6 antibody. Cells were lysed with PBS/1% Triton X-100, and the lysate was plated on LB agar. The number of CFU was counted after 24 h incubation. Each dot represents six independent replicates (n = 6 per group). Data are represented as mean ± SD. **$p < 0.01$. *p* values were calculated by one way analysis of variance.

## Administration of Gas6 prevents systemic infection by orally infecting *K. pneumoniae* in elderly mice

Based on our findings, we hypothesized that the administration of Gas6 would inhibit bacterial invasion of epithelial cells. To test this hypothesis, recombinant Gas6 protein was exposed to the apical or basolateral sides of Caco-2 cells grown in a Transwell plate for 18 h (**Fig 6A**). Regardless of the side exposed to Gas6 recombinant protein, the expression of Axl, ZO-1, and occludin increased in Caco-2 cells stimulated with Gas6 (**Figs 6B and S7A**). Next, we added Gas6 recombinant protein (1 µg) to the apical side of Caco-2 cells grown in Transwell co-culture systems for 3 h prior to *K. pneumoniae* infection. After 1.5 h of infection with *K. pneumoniae*, the cells were incubated for 6 h with DMEM containing 100 µg/mL gentamycin and 1 µg Gas6 recombinant protein to kill extracellular bacteria. Administration of the Gas6 recombinant protein increased the expression of Axl, ZO-1, and occludin in Caco-2 cells in the absence of *K. pneumoniae* (**Figs 6C, lanes 1 and 2, and S7B**). Increased expression of these proteins induced by the Gas6 recombinant protein was also detected in Caco-2 cells infected with *K. pneumoniae* (**Figs 6C, lanes 3 and 5 and S7B**). Immunostaining analysis revealed that the expression of ZO-1 and occludin (but not E-cadherin) by *K. pneumoniae*-infected Caco-2 cells in the absence of RAW264.7 macrophages increased significantly after treatment with Gas6 recombinant protein (**Fig 6D and 6E**). Also, in uninfected Caco-2 cells, treatment of Gas6 recombinant protein induced ZO-1 and occludin expression, but not E-cadherin (**S8A and S8B Fig**). Furthermore, treatment with Gas6 recombinant protein significantly decreased the signals of *K. pneumoniae* immunostaining within Caco-2 cells (**Fig 6F and 6G**). These findings show that administration of Gas6 instead of RAW264.7 macrophages can repress bacterial invasion of intestinal epithelial cells by enhancing tight junction barriers.

Next, we investigated whether Gas6 administration protects elderly mice against *K. pneumoniae* infection. Prior to bacterial infection, 57-week-old mice received three intraperitoneal injections of Gas6 recombinant protein (125 µg protein/kg) every 24 h (**Fig 7A**). There was no significant difference between the survival influences of 15-week-old mice and Gas6-treated 57-week-old mice upon infection with *K. pneumoniae*, indicating that Gas6 recombinant proteins protect the elderly host against *K. pneumoniae* infectious disease (**Fig 7B**). The desquamation area of the cecal epithelium observed in *K. pneumoniae*-infected 57-week-old mice was barely detectable in Gas6-treated 57-week-old mice (**S9A and S9B Fig**). While the number of bacteria invading the cecal mucosa and translocating to the liver was significantly higher in 57-week-old mice than in 15-week-old mice, the increase was significantly attenuated by the administration of Gas6 recombinant protein (**Fig 7C**). There was no difference of the bacterial number in the cecal contents between 15-week- and 57-week-old mice, suggesting that the aging does not affect the colonization of the bacteria in antibiotics-administered mice (**Fig 7C**). In addition, the administration of Gas6 did not reduce the infected bacterial number in the cecal contents, indicating that Gas6 does not directly reduce the infected bacterial load

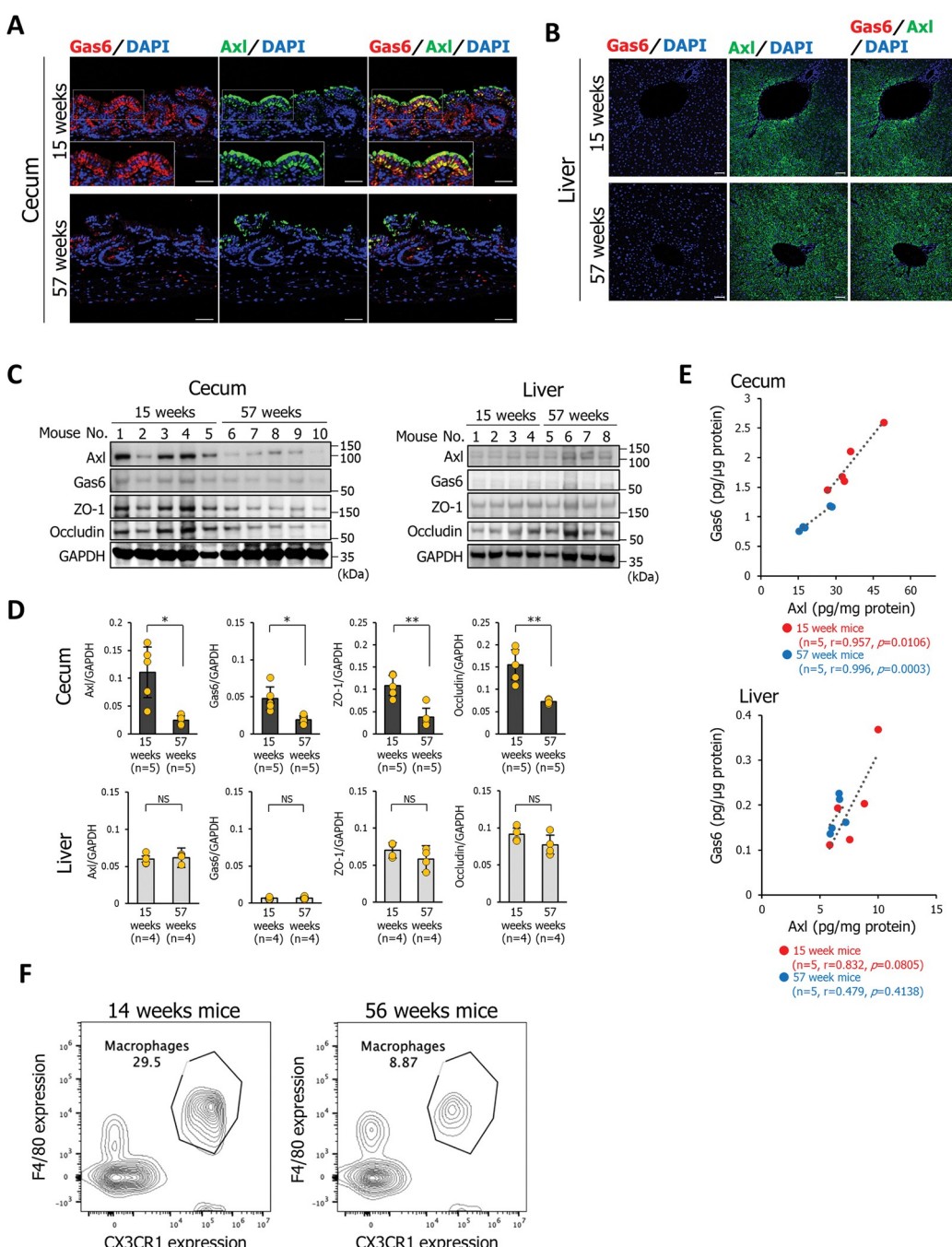

**Fig 5. Gas6 and Axl expression in the cecal mucosa during *K. pneumoniae* infection decreases with aging. (A and B)** Sections of cecal mucosa (A) or liver (B) were from mice aged 15 or 57 weeks at 2 days after infection with *K. pneumoniae* ATCC43816 pmCherry and immunostained with an anti-Gas6 antibody and an anti-Axl antibody. Each image has six independent replicates. Scale bar = 50 μm. **(C)** Detection of Axl, Gas6, ZO-1, and occludin by western blotting. Cecum and liver tissues were collected, homogenized, and analyzed using an anti-Axl antibody, an anti-Gas6 antibody, an anti-ZO-1 antibody, or an anti-occludin antibody. Each lane of western blotting images represents the protein from an individual mouse (mouse number = 5 (cecum) or 4 (liver) per group). Western blotting analysis represents three independent replicates. **(D)** Western blotting signal intensities were analyzed by ImageJ software. Each dot represents the value collected from an individual mouse (mouse number = 5 (cecum) or 4 (liver) per group). Data are presented as the mean ± SD. NS: not significant, *$p < 0.05$, **$p < 0.01$. $p$ values were calculated by the Student's $t$ test. **(E)** Linear correlation between Gas6 expression and Axl expression in the cecum or liver of mice aged 15 (red circles) or 57 (blue circles) weeks infected with *K. pneumoniae* ATCC43816 pmCherry. Each dot represents an ELISA measured value of each protein sample collected from an individual mouse (mouse number = 5 per group). $r > 0.70$ and $p < 0.05$. **(F)** The population of CD11b+F4/80

+ macrophages in the intestinal mucosa of young (14-week-old) and old (56-week-old) mice was examined by staining with anti-F4/80 and anti-CD11b antibodies. The percentage of F4/80-positive/CX3CR1-negative cells in 14-week-old and 56-week-old mice was 29.5% and 8.87%, respectively. Data are representative of four mice per group.

(**Fig 7C**). To next examine whether administration of Gas6 restores the number of intestinal mucosal macrophages in 57-week-old mice, the expression levels of F4/80 were measured by western blotting and immunofluorescence analysis. The expression levels and immunostaining signals of F4/80 in the cecal mucosa of *K. pneumoniae*-infected 57-week-old mice were significantly decreased compared with that of 15-week-old mice (**Figs 7D, 7E, S9C and S9D**). The reduced F4/80 expression levels of *K. pneumoniae*-infected 57-week-old mice were not restored by Gas6 administration (**Figs 7D 7E, S9C and S9D**). We then analyzed the localization of Gas6 recombinant protein in the cecal mucosa of *K. pneumoniae*-infected mice administered with Gas6. While colocalization of Gas6 and Axl in the cecal epithelium of 15-week-old mice was observed, the staining signals of Gas6 were scarcely detected in 57-week-old mice (**Fig 7F and 7G**). Administration of Gas6 recombinant protein in the 57-week-old mice restored the staining levels of Gas6 on the cecal epithelium and the colocalization of Gas6 and Axl was also detected (**Fig 7F and 7G**). Further, the expression levels of Axl, ZO-1, and occludin in the cecal mucosa of Gas6-treated 57-week-old mice infected with *K. pneumoniae* were also restored to levels similar to those in 15-week-old mice (**Fig 7H and 7I**). These findings suggest that Gas6 recombinant protein enhances the tight-junction barriers in the intestinal epithelium by directly stimulating Gas6/Axl signals, leading to the protective effect being exerted against *K. pneumoniae* infection in elderly hosts.

## Discussion

*K. pneumoniae* is a potentially pathogenic gut pathobiont that can become pathogenic under specific genetic, environmental, or immunocompromised conditions [12,13]. However, the environmental or immunological conditions that transform *K. pneumoniae* from a gut symbiont to a pathogenic organism remain unclear. Here, we showed that reduced Gas6/Axl signals with aging in the intestinal mucosa induce the invasion of *K. pneumoniae* into the intestinal epithelium, and these bacteria then easily translocate to the liver.

*K. pneumoniae* ATCC43816 is a well-studied hypervirulent laboratory strain which has K2 serotype capsules and can cause an acute respiratory disease in animal models [14]. Recently, genomic sequencing analysis revealed that *K. pneumoniae* ATCC43816 possess the known virulence determinant for a siderophore-dependent iron uptake system (yersiniabactin), a Type 4 secretion system, and a CRISPR system, which are absent from non-hypervirulent strains [14]. In this study, we show that Gas6 inhibits the invasion of *K. pneumoniae* ATCC43816 to the intestinal epithelium by enhancing tight-junction barriers, contributing to the prevention of the bacterial translocation. Since hypervirulent strains have the ability to infect young, healthy individuals, the activity of Gas6 in preventing the bacterial translocation of hypervirulent strains is beneficial information to prevent *K. pneumoniae* infection.

*K. pneumoniae* is reported to invade human epithelial cell lines. The invasion rates among each clinically isolated strain range from 10.1–48.6%, indicating that invasive abilities are differ among the bacterial strains [15,16]. The difference in invasiveness is reported to not be dependent on capsule type, suggesting that invasion abilities are mediated by multifactorial process and that capsule alone is insufficient to account for differences in pathogenicity between strains [15]. Pan *et al.* also reported that additional genetic loci, including genes for lipopolysaccharides (LPS), the ClpX protease, and carnitine metabolism, contribute to the virulence of *K. pneumoniae* [17]. From these observations, it was thought that virulence factors of *K.*

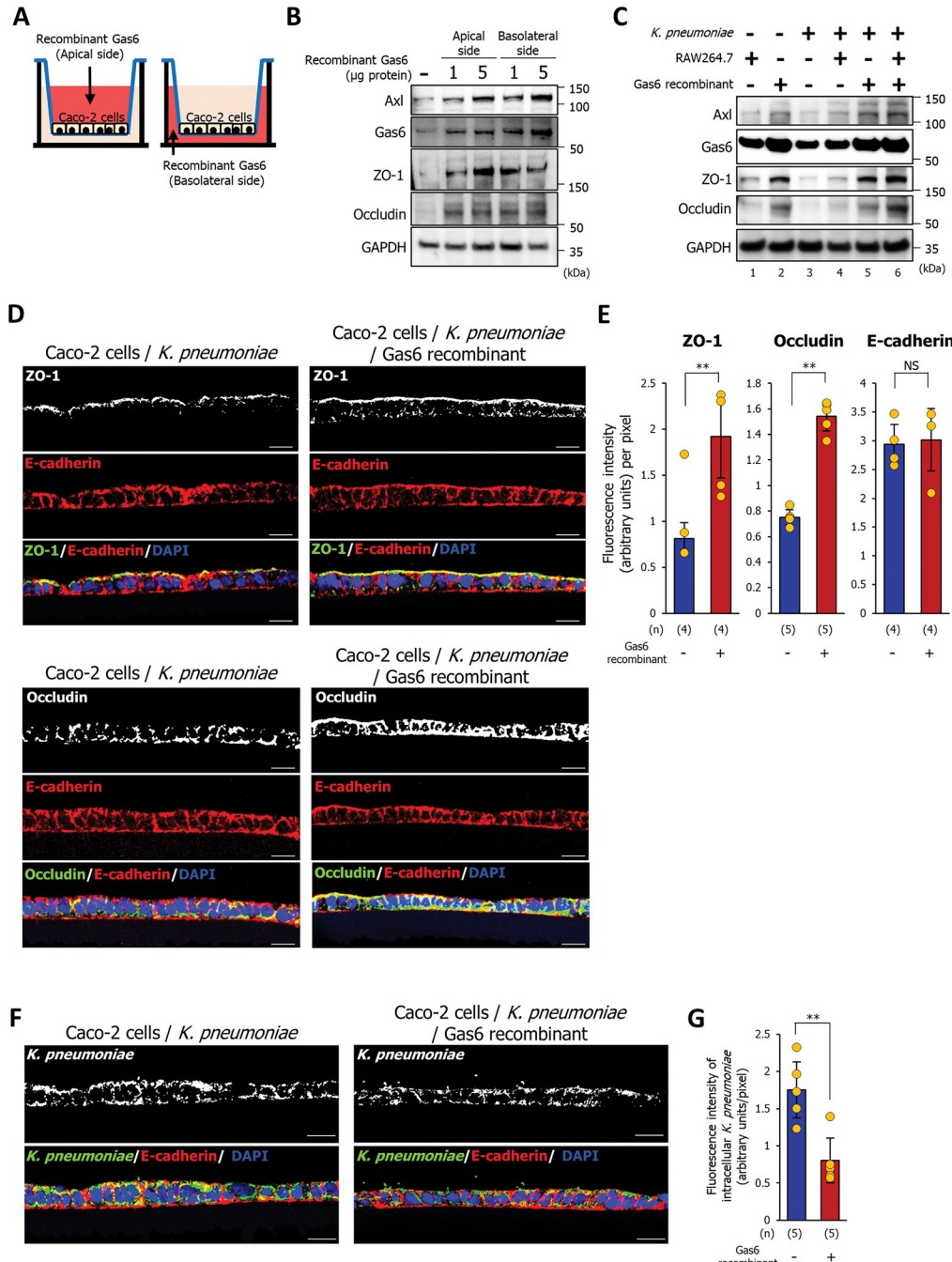

**Fig 6. Gas6 represses the invasion of *K. pneumoniae* into Caco-2 cells by increasing the expression of ZO-1 and occludin. (A)** Administration of Gas6 recombinant protein to Caco-2 cells to analyze the expression of Axl, ZO-1, and occludin. Addition of Gas6 recombinant protein to the apical surface (left scheme) or the basolateral side (right scheme) of Caco-2 cells. **(B)** Western blot analysis to detect Axl, Gas6, ZO-1, and occludin in Caco-2 cells treated with human Gas6 recombinant protein from the apical or basolateral sides. Each western blotting image represents three independent replicates (n = 3). **(C)** Western blot analysis was performed to detect Axl, Gas6, ZO-1, and occludin. Gas6 recombinant protein (1 μg) was added to the apical side of Caco-2 cells grown in a Transwell co-culture system for 3 h prior to *K. pneumoniae* infection. Each western blotting image represents three independent replicates (n = 3). **(D)** Gas6 recombinant protein (1 μg) was added to Caco-2 cells grown on the Transwell insert for 3 h prior to *K. pneumoniae* infection. The cells were immunostained with an anti-ZO-1 antibody or an anti-occludin antibody, and an anti-E-cadherin antibody. Each image has four independent replicates. Scale bar = 20 μm. **(E)** Fluorescence intensity per pixel of anti-ZO-1 antibody, an anti-occludin antibody, and anti-E-cadherin antibody were measured by ImageJ analysis software. Each dot represents four independent replicates (n = 4 or 5 per group). Data are presented as the

mean ± SD. NS: not significant, **$p < 0.01$. $p$ values were calculated by the Student's $t$ test. **(F)** Gas6 recombinant protein (1 µg) was added to Caco-2 cells grown on the Transwell insert for 3 h prior to *K. pneumoniae* infection. The cells were immunostained with an anti-*Klebsiella pneumoniae* antibody and an anti-E-cadherin antibody. Each image has four independent replicates. Scale bar = 20 µm. **(G)** Fluorescence intensity per pixel of anti-*Klebsiella pneumoniae* antibody were measured by the ImageJ analysis software. Each dot represents five independent replicates (n = 5 per group). Data are presented as the mean ± SD. **$p < 0.01$. $p$ values were calculated by the Student's $t$ test.

*pneumoniae* are diverse and it may be difficult to identify the factors which determined the difference in virulence among the bacterial strains. The protective ability of Gas6 to the invasion of *K. pneumoniae* depends on the Gas6/Axl signaling in epithelial cells, and therefore Gas6 can be inferred to prevent the bacterial invasion regardless of the bacterial strain. Further studies are needed to determine whether Gas6 also prevents the bacterial translocation from the gastrointestinal tract against non-hypervirulent *K. pneumoniae* strains, which are particularly prevalent in the elderly, by analyzing the inducibility of Gas6 secretion from macrophages by several strains.

Host defense mechanisms are maintained by a variety immune-related factors influenced and regulated by the gut microbiota [18]. The influence of gut microbiota to the host not only fosters development and aids digestion but also directly protects the host from infection by pathogens, a function referred to colonization resistance to pathogens [19]. We recently reported that short-chain fatty acids (SCFAs), which are metabolites of gut microbiota, promoted inflammasome activation in macrophages by binding to its apoptosis-associated speck-like protein (ASC), leading to the enhanced elimination of *Salmonella enterica* serovar Typhimurium (*S. Typhimurium*) within macrophages [20]. Dr. Sequeira *et al.* reported that *Bacteroidetes* specifically protects against intestinal colonization by *K. pneumoniae* through IL-36 signaling [21]. In this study, to exclude the effects of colonization resistance by microbiota on the *K. pneumoniae*, antibiotic-treated mice were used. Rakoff-Nahoum and colleagues reported that the combination therapy of four antibiotics (ampicillin, metronidazole, neomycin, and vancomycin) for mice was used for complete depletion of gut commensal bacteria [22]. This combination of antibiotics significantly decreases the concentrations of SCFAs (propionate and butyrate) in the cecal lumen [20]. Using mice treated with a combination of four antibiotics, induction of Gas6 expression was detected in the cecal mucosa of 15-week-old mice infected with *K. pneumoniae*, suggesting that Gas6 expression, under *K. pneumoniae* infection, is the event that independently affects the composition of gut microbiota (**Fig 5A, 5C and 5D**). Our results reveal a novel host-defense mechanisms in which Gas6 secreted by macrophages protects against *K. pneumoniae* infection, in addition to the defensive mechanisms by *Bacteroidetes*.

Further studies using germ-free mice would be useful to exclude the four other potential factors which could influence the protective activity of Gas6: (1) Effect of changing the abundance and composition of relevant immune cell subsets, such as reduced secretion of antimicrobial peptides and changes in T helper cell populations, by administration of antibiotics [23], (2) Effect of the difference in composition of the immune system influenced by early encounters with microorganisms [24], (3) Effect of the immune-systems modulated by early priming for *K. pneumoniae*, (4) The possibility that gut microbiota in antibiotic-treated mice were not completely eradicated. From the study using germ-free mice, it is expected that detailed information on the benefit of Gas6 for the protective and/or therapeutic activity against *K. pneumoniae* infection in the elderly would be revealed.

*K. pneumoniae* caused desquamation of the cecal intestinal epithelium of 57-week-old mice and Caco-2 cells (**S1B and S2 Figs**). According to Nakamoto *et al.* [25], *K. pneumoniae* induces epithelial pore formation, which disrupts the intestinal barrier and allows bacterial

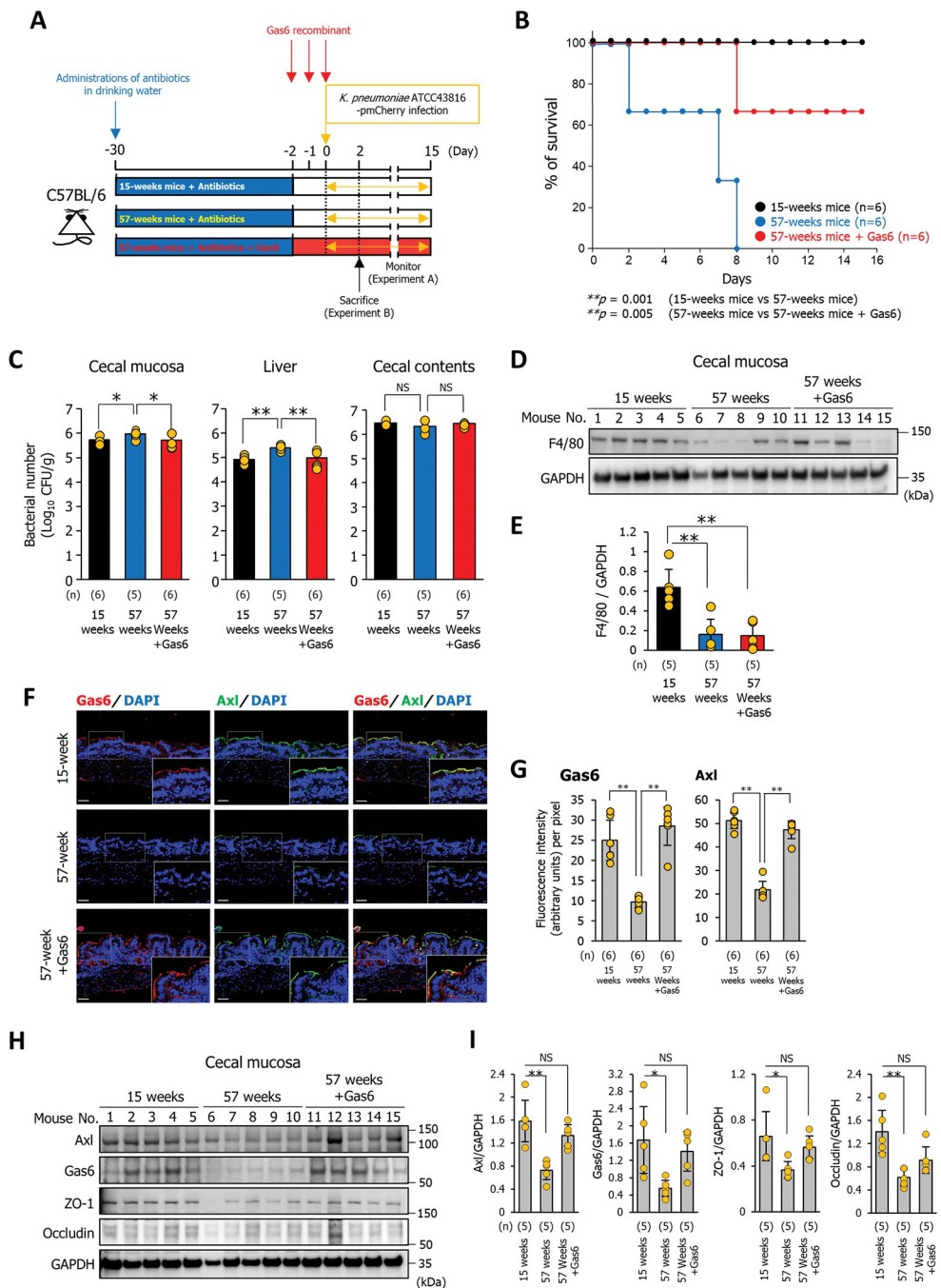

**Fig 7. Gas6 prevents systemic infection by orally infecting *K. pneumoniae* in elderly mice. (A)** Treatment scheme used to analyze the effect of Gas6 recombinant protein on the susceptibility of 57-week-old mice to infection by *K. pneumoniae*. Antibiotics were administered 4 weeks before administration of Gas6 recombinant protein. Gas6 recombinant protein (125 μg protein/kg) was administered intraperitoneally to 57-week-old mice three times every 24 h prior to bacterial infection. Survival was monitored daily (Experiment A). At 2 days post-infection, the mice were sacrificed, and the cecum and liver were harvested (Experiment B). **(B)** Effect of Gas6 recombinant protein on survival of 57-week-old mice infected with *K. pneumoniae* ATCC43816 pmCherry. Each mouse was orally inoculated with *K. pneumoniae* ATCC43816 pmCherry ($5 \times 10^7$ bacteria). Each dot was represented from an individual mouse (n = 6 per group). *p* values were determined using the log-rank test. **(C)** Bacterial counts in cecal mucosa, liver, and cecal contents were determined 2 days post-infection. Each tissue and cecal contents were homogenized in PBS. The homogenates were plated on LB agar containing 400 μg/mL ampicillin and the number of CFU was counted. Each dot represents the value from an individual mouse (n = 5 or 6 per group). Data are presented as the mean ± SD. *$p < 0.05$, **$p < 0.01$. *p* values were calculated by one way analysis of variance. **(D)** Detection of F4/80 by western blotting. Cecal mucosa were

collected, homogenized, and analyzed by western blotting with anti-F4/80 antibodies. Each lane of western blotting images represents the protein from an individual mouse (mouse number = 5 per group). Western blotting image represents three independent replicates. **(E)** Western blotting signal intensities were analyzed by ImageJ software. Each dot represents the protein sample collected from an individual mouse (mouse number = 5 per group). Data are presented as the mean ± SD. ** $p < 0.01$. $p$ values were calculated by one way analysis of variance. **(F)** Sections of cecal mucosa were from mice aged 15-week, 57-week, or Gas6 administered 57-week-old mice at 2 days after infection with *K. pneumoniae* ATCC43816 pmCherry and immunostained with an anti-Gas6 antibody and an anti-Axl antibody. Each image has six independent replicates. Scale bar = 50 μm. **(G)** Fluorescence intensity was analyzed by ImageJ software. Each dot represents the value from an individual mouse (n = 6 per group). Data are presented as the mean ± SD. ** $p < 0.01$. $p$ values were calculated by one way analysis of variance. **(H)** Detection of Axl, Gas6, ZO-1, and occludin by western blotting. Cecal mucosa were collected, homogenized, and analyzed by western blotting with anti-Axl, anti-Gas6, anti-ZO-1, or anti-occludin antibodies. Each lane of western blotting images represents the protein from an individual mouse (mouse number = 5 per group). Western blotting image represents three independent replicates. **(I)** Western blotting signal intensities were analyzed by ImageJ software. Each dot represents the value collected from an individual mouse (mouse number = 5 per group). Data are presented as the mean ± SD. NS: not significant, * $p < 0.05$, ** $p < 0.01$. $p$ values were calculated by one way analysis of variance.

translocation. Furthermore, the pore-forming ability of *K. pneumoniae* is considered strain-specific [25]. Recently, hypervirulent strains of *K. pneumoniae* were isolated from patients with liver abscesses; these strains exert pathogenic effects even in immune-competent healthy individuals [26]. Conserved genes responsible for pathogenic pore-forming ability have not been identified; therefore, it is unclear whether the hypervirulent strains and the ATCC43816 strain used in this study encode pathogenic genes related to pore-forming capacity. Although further investigations, including the identification of pore-forming toxins, are needed to identify the precise mechanisms by which *K. pneumoniae* induces epithelial injury, the current findings indicate that age-related changes in host-bacterial interactions play a significant role in the induction of *K. pneumoniae*-mediated epithelial cell injury and subsequent bacterial translocation to the liver.

As shown in Fig 3, *K. pneumoniae* is thought to produce a Gas6-inducing factor. In Caco-2 cells co-cultured with RAW 264.7 macrophages primed by *E. coli*-derived LPS without *K. pneumoniae* infection, Gas6/Axl signaling-dependent enhancement of ZO-1 and occludin expression was not observed (**S3 and S5 Figs**). Thus, the Gas6-inducing factor is thought to be a specific molecule produced by *K. pneumoniae*. Interestingly, since the activity of *K. pneumoniae* to induce Gas6 secretion was not inactivated by heat treatment, it is assumed unlikely that the Gas6-inducing factor is a protein molecule, but rather some specific-LPS or capsule type produced by *K. pneumoniae*. The identification and purification of the Gas6-inducing factor contribute to elucidating the detailed mechanism of Gas6 secretion by macrophages.

Organoid cultures are known to be excellent platforms to examine host–microbe interaction mechanisms and triple co-culture systems of organoid, immune cells, and bacteria are reported to be a futuristic next step [27]. By utilizing a triple co-culture system of intestinal organoids, BMDMs, and *K. pneumoniae*, it is expected that the detailed mechanisms of the interaction between *K. pneumoniae* and host epithelial cells will be revealed.

We found a significant linear correlation between the expression levels of Gas6 and Axl in the cecal mucosa (**Fig 5D**). In contrast, this linear correlation and reduced expression of tight junction proteins were not observed in the liver (**Fig 5D**). These results suggest that the mechanism that regulates Gas6/Axl signaling differs between the cecal mucosa and the liver. Liver-resident macrophages (Kupffer cells) are estimated to not secrete Gas6 (or very low amounts); in fact, Gas6 expression in the liver was barely detectable by immunostaining (**Fig 5B and 5C**). Additionally, it is also conceivable that *K. pneumoniae* translocated to the liver may not release a factor that increases Gas6 production by Kupffer cells. Our findings indicate that macrophage-dependent Gas6/Axl signaling in the intestinal epithelium is essential for maintaining the intestinal mucosal barrier.

Several reports show the immune system deteriorates with age, immunosenescence was revealed to contribute to not only adaptive immunity dysfunction but also altered innate immunity [28,29]. Our results show that in the antibiotic-treated mice, the colonization resistance to orally infected *K. pneumoniae* into the cecal lumen is not affected by aging (**Fig 7C**). Elucidating the dysfunctional mechanisms of immune systems that contribute to increased susceptibility to *K. pneumoniae* is important and necessary to understand *K. pneumoniae* infection in the elderly patients. Although the present study shows the protective effects of Gas6 secreted from macrophages against *K. pneumoniae* infection in elderly mice, further studies are needed to determine the differences in the immune system of young and elderly mice which contribute to the susceptibility to *K. pneumoniae*.

Previously, *K. pneumoniae* has been known as the "gut pathobiont" that causes serious infections, primarily in immunocompromised individuals. However, multidrug-resistant and hyper-virulent strains of *K. pneumoniae* have been identified, which are capable of causing untreatable infections in healthy individuals [30–33]. Thus, a detailed understanding of the biology underlying infectious behavior is necessary to develop new treatment and prevention strategies. The present study shows that the administration of exogenous Gas6 protects elderly mice against *K. pneumoniae* infection by preventing reductions in ZO-1 and occludin expression in the intestinal mucosa (**Fig 7**). Axl is expressed not only on the apical but also on the basolateral cell surface membrane [34]. *In vitro* experiments also showed that the addition of exogenous Gas6 to the apical or basolateral side induced ZO-1 and occludin expression (**Fig 6B**).

Antibiotic-treated mice were used for the *in vivo* infection model in this study. Administration of antibiotics is reported to change the abundance and composition of immune cell subsets [23]. Therefore, to determine whether Gas6 has an effect, even in a healthy homeostatic environment in the gut, we also need to examine the protective ability of exogenous Gas6 to SPF-mice infected with *K. pneumoniae*. Taken together, these results indicate that different routes of administration can be considered when using a Gas6 or Gas6 mimic compound to modulate Gas6/Axl signaling in the intestinal epithelium. Our findings provide new insights into strategies to combat *K. pneumoniae* infectious diseases in the elderly.

## Methods

### Ethics statement

All animal experiments were approved by the Keio University (Tokyo, Japan) Animal Research Committee (no. 19048) and the Tokai University (Kanagawa, Japan) Animal Research Committee (no. 222002) and were conducted in accordance with the "Act on Welfare and Management of Animals of Japan," "Standards relating to the Care and Keeping and Reducing Pain of Laboratory Animals," "Standards relating to the Methods of Destruction of Animals," "Guidelines for Proper Conduct of Animal Experiments," and "Fundamental Guidelines for Proper Conduct of Animal Experiments."

### Reagents and antibodies

An Axl inhibitor (R428) (Abcam, Cambridge, UK, cat# ab141364), recombinant human Gas6 protein (R&D Systems, Minneapolis, MN, USA, cat# 885-GSB), recombinant mouse Gas6 protein (R&D Systems, cat# 986-GS) (used in administration experiments in mice), and an anti-Gas6 antibody (R&D Systems, cat# AF986) were used as Gas6/Axl signal modulators. Rhodamine phalloidin (Invitrogen, Waltham, MA, USA, R415) was used to stain the polymerized actin. The following antibodies were used for immunostaining: anti-*Klebsiella pneumoniae* (Thermo Fisher Scientific, Cleveland, OH, USA, cat# PA1-7226), anti-E-cadherin (BD Biosciences, San Jose, CA, USA, cat# 610181), anti-ZO-1 (Invitrogen, cat# 61–7300), anti-occludin

(Invitrogen, cat# 711500), anti-Axl (GeneTex Inc., Irvine, CA, USA, cat# GTX129407), anti-Gas6 (R&D Systems, cat# AF986), anti-F4/80 (Bio-Rad, Hercules, CA, USA, cat# MCA497R), and anti-mCherry (Invitrogen, cat# M11217). The following antibodies were used for western blotting: anti-Axl (GeneTex Inc., cat# GTX129407), anti-Gas6 (R&D Systems, cat# AF986), anti-ZO-1 (Invitrogen, cat# 61–7300), anti-occludin (Invitrogen, cat# 711500), anti-F4/80 (Bio-Rad, cat# MCA497R), and anti-GAPDH (Cell Signaling Technologies, Danvers, MA, USA, cat# 2118S). The following antibodies were used for flow cytometry analysis: FITC-conjugated anti-CD19 (6D5) (BioLegend, San Diego, CA, USA, cat# 115505), APC-conjugated anti-CD11b (M1/70) (BioLegend, cat# 101211), PerCP-Cy5.5-conjugated anti-CD3 (17A2) (BioLegend, cat# 100217), and PE-Cy7-conjugated anti-F4/80 (BM8) (BioLegend, cat# 123113).

## Cell and bacterial culture

Caco-2 cells, which were purchased from the European Collection of Cell Cultures (ECACC 86010202), were cultured in DMEM (Gibco, Waltham, MA, USA, cat# 11965092) supplemented with 10% fetal bovine serum (FBS), 1% non-essential amino acids (Gibco, cat# 11140–050), and 2 mM L-glutamine (Gibco, cat# 25030–081). RAW264.7 cells purchased from the American Type Culture Collection (ATCC TIB-71) were maintained in DMEM supplemented with 10% FBS. BMDMs were prepared by flushing bone marrow from femurs and tibiae of 20-week-old mice with DMEM, and were cultured in DMEM supplemented with 10% FBS and 20 ng/mL recombinant mouse macrophage-colony stimulating factor (PeproTech, Rocky Hill, NJ, USA, cat. no. 315–02) for 7 days [20]. *K. pneumoniae* ATCC43816 was cultured overnight at 37˚C on Luria–Bertani (LB) agar (Nacalai Tesque Inc., Kyoto, Japan, cat# 20067–85). Bacterial counts were determined by measuring the optical density of bacterial suspensions at 550 nm.

## Mice

All the male mice were bred and maintained under specific pathogen-free (SPF) conditions. Six-week-old C57BL/6J male mice were purchased from SLC Japan, Inc. (Shizuoka, Japan) or CLEA Japan, Inc. (Osaka, Japan) and used in experiments at 14 or 15 weeks of age. C57BL/6J mice aged 48–55 weeks were purchased from SLC Japan or CLEA Japan and were used at 57 weeks of age. A previous study reported that the administration of four antibiotics (ampicillin, metronidazole, neomycin, and vancomycin) to mice excluded endogenous short-chain fatty acids (SCFAs) by completely eradicating gut commensal bacteria [20]. To exclude the effects of the gut microbiota, 11 or 53-week-old mice were provided with drinking water containing 1 g/L ampicillin (Sigma-Aldrich, St. Louis, MO, USA, cat# A0166), 1 g/L metronidazole (Sigma-Aldrich, cat# M1547), 1 g/L neomycin (Sigma-Aldrich, cat# N1876), and 0.5 g/L vancomycin (Wako, Osaka, Japan, cat# 222–01303) for 4 weeks [20,22,35]. Mice that lost more than 20% of their body weight because of their refusal to drink the antibiotic cocktail were excluded from the experiments in accordance with the guidelines of the Keio University Animal Research Committee (no. 19048) or the Tokai University Animal Research Committee (no. 222002). In animal experiments, two independent experiments were performed on different days. The first experiment was performed to collect preliminary data and the second experiment was performed for data collection. The number of representative image replicates, and the number of each sample, is described within the relevant Figure legend. Blind analysis of the data was ensured by providing investigators with numbered images. Image data were saved as TIFF files. Images for which the analysis by ImageJ software showed average value results were selected as representative images.

## Isolation of murine macrophages and Flow cytometry

Intestinal lamina propria cells were isolated from 14 or 56-week-old mice, as described previously with slight changes [36]. The harvested intestine was cut into 10-mm long segments, opened longitudinally, and washed in phosphate-buffered saline (PBS). The washed segments were treated for 10 min at room temperature with 30 mM EDTA. These intestinal pieces were washed thoroughly with PBS and digested for 30 min at 37˚C with 0.5 mg/mL collagenase D (Merck/Sigma-Aldrich, cat# 11088858001) in RPMI medium (Gibco, cat# 11875093) supplemented with 2% fetal calf serum (FCS). After repeated pipetting, the supernatant was collected after 2 min and passed through a cell strainer (70 μm). The remaining tissue fragments were subjected to a second collagenase digestion, and these steps were repeated until no gross tissue fragments were visible. The digested tissues were washed with 10 mL of PBS, resuspended in 5 mL of 40% Percoll (GE Healthcare, Chicago, IL, USA), and underlaid with 5 mL of 80% Percoll (GE Healthcare) in a 15 mL tube. The Percoll gradient separation was performed by centrifugation at $800 \times g$ for 15 min at room temperature. Lymphocytes were collected from the interface of the Percoll gradient and washed with RPMI1640 containing 10% FBS. For cell surface staining, $2 \times 10^6$ cells were stained with fluorescence-conjugated anti-F4/80, anti-CD11b, anti-CD19, and anti-CD3e antibodies for 35 min on ice in the presence of an anti-CD16/32 (2.4G2) antibody for blocking. The cells were analyzed using a Cytoflex flow cytometer (Beckman Coulter, Tokyo, Japan). The collected data were analyzed using Flowjo software (Tree Star, Ashland, Oregon, USA). Antibody dilutions were 200-fold for surface staining.

## *K. pneumoniae* infection of mice

To distinguish between intestinal commensal *K. pneumoniae* strains and orally infected *K. pneumoniae* ATCC43816 strains, a pmCherry plasmid (Clontech, Palo Alto, CA, USA, cat #632522) was electroporated into *K. pneumoniae* ATCC43816 using a MicroPulser (Bio-Rad, Hercules, CA, USA). SPF mice aged 11 or 53 weeks were provided with drinking water containing four antibiotics (ampicillin, metronidazole, neomycin, and vancomycin) for 4 weeks prior to *K. pneumoniae* ATCC43816 pmCherry infection. Administration of these antibiotics was stopped prior to bacterial infection. Mice were orally inoculated with 200 μL LB broth containing *K. pneumoniae* ATCC43816 pmCherry ($5 \times 10^7$ bacteria) using a stainless-steel feeding needle fitted to a 1.0 mL syringe and their survival was monitored daily. To assess the number of bacteria in cecal mucosa, cecal contents and liver, *K. pneumoniae* ATCC43816 pmCherry-infected mice were sacrificed 2 days' post infection, and tissues and cecal contents were harvested, suspended in PBS, and homogenized with a Qiagen TissueLyser (Qiagen, Hilden, Germany). Serial dilutions of the homogenates were plated on LB agar containing 50 μg/mL ampicillin, and colony-forming units (CFUs) were counted after 24 h of incubation.

Sections (4 μm) of the cecum or liver collected from mice infected with *K. pneumoniae* ATCC43816 pmCherry ($5 \times 10^7$ bacteria) for 2 days were fixed overnight in 10% formalin neutral buffer solution and embedded in paraffin. The sections were then deparaffinized, and stained with H&E. Desquamation of the epithelial cells or edema of the cecal submucosa was examined using an OPTIKA B-290TB Digital Microscope (Optika Microscope, Milano, Italy). The length of desquamated cecal epithelial cells on H&E-stained specimens was measured using the ImageJ analysis software (National Institute of Health). For immunohistochemistry analysis, tissue sections were treated as described above and incubated overnight at 4˚C with antibodies specific for mCherry (Invitrogen), Gas6 (R&D Systems), Axl (GeneTex Inc.), or F4/80 (Bio-Rad) followed by incubation for 1 h with DAPI and Alexa Fluor 488-conjugated anti-rat, anti-rabbit, or Alexa Fluor 568-conjugated anti-mouse IgG secondary antibodies. Fluorescence images were obtained using a LSM700 confocal microscope (Carl Zeiss, Oberkochen,

Germany). Fluorescence intensity of immunostaining was measured as intensity values per pixel using ImageJ analysis software (National Institute of Health, Bethesda, MD, USA). For western blotting, cecum or liver samples collected from each mouse were homogenized in RIPA buffer containing protease inhibitors. Total protein (10 μg/lane) was separated on 10% Bis-Tris Plus gels (Thermo Fisher Scientific, NW00107BOX) and transferred to polyvinylidene difluoride membranes (Amersham, Braunschweig, Germany, cat# 10600122). An anti-GAPDH antibody (Cell Signaling Technologies) was used as a loading control. Western blotting signal intensity of each band was measured using ImageJ software. The number of representative image replicates and the number of each sample is described within Figure legend and Figure. Image data were saved as TIFF files, and images for which the image analysis by ImageJ analysis software results showed an average value were selected as representative images.

### *In vitro K. pneumoniae* infection model of Caco-2 cells grown in Transwell culture systems

Caco-2 cells ($1 \times 10^6$ cells/well) were seeded onto 6-Transwell insert culture plates (0.4 μm pore size) (Corning, Lowell, MA, USA, cat# 3412). RAW264.7 macrophages or BMDMs were seeded onto 6-well flat-bottom cell culture plates at $1 \times 10^5$ cells per well. To establish a co-culture system for Caco-2 cells and RAW264.7 macrophages or BMDMs, inserts on which Caco-2 cells were grown were placed in the wells of culture plates containing RAW264.7 macrophages or BMDMs. RAW264.7 macrophages and BMDMs were primed for 6 h with 1 μg/mL LPS prior to *K. pneumoniae* infection. The apical side of Caco-2 cells grown on Transwell inserts were exposed for 3 h to 20 nM Axl inhibitor (R428) (Abcam), 1 μg of anti-Gas6 antibody (R&D Systems), or 1 μg of human recombinant Gas6 protein (R&D Systems) prior to *K. pneumoniae* infection. *K. pneumoniae* was resuspended in DMEM and incubated for 1.5 h with Caco-2 cells at a multiplicity of infection of 50. The cells were then washed three times with PBS and incubated for 6 h in DMEM containing 400 μg/mL gentamycin. To measure the number of surviving intracellular cells, cells were washed with PBS and lysed with PBS/1% Triton X-100, followed by plating on LB agar. CFUs were counted after 24 h of incubation. For immunopathological analysis, the Transwell membrane was fixed overnight in 4% paraformaldehyde and embedded in paraffin. Sections (4 μm) were deparaffinized, stained with H&E, and incubated overnight at 4°C with antibodies specific for E-cadherin (BD Biosciences, Franklin Lakes, NJ, USA), *K. pneumoniae* (Thermo Fisher Scientific), ZO-1 (Invitrogen), occludin (Invitrogen), Gas6 (R&D Systems), or Axl (GeneTex Inc.). The sections were then incubated for 1 h with Alexa Fluor 488-conjugated anti-rabbit or Alexa Fluor 568-conjugated anti-mouse IgG. Fluorescence images were obtained using an LSM710 or 700 confocal microscopes (Carl Zeiss, Oberkochen, Germany). Quantification of *K. pneumoniae*, E-cadherin, ZO-1, and occludin staining was performed using the ImageJ analysis software (National Institutes of Health). For western blotting, cells were collected from the Transwell membrane using Cell Scraper and treated with RIPA buffer containing protease inhibitors. Total protein was separated (10 μg protein/lane) on 10% Bis-Tris Plus gels (Thermo Fisher Scientific). Caco-2 cells grown on Transwell inserts were incubated with or without *K. pneumoniae* for 1.5 h and then incubated for 6 h in DMEM containing 400 μg/mL gentamycin. The culture medium was collected for cytokine array analysis (QAM-CAA-4000; RayBiotech, Peachtree Corners, GA, USA).

### Measurement of Axl and Gas6

Caco-2 and RAW264.7 macrophages ($1 \times 10^6$ cells per well) were cultured in 6-Transwell insert culture plates (Corning) or 6-well flat-bottom cell culture plates, respectively. *K.*

*pneumoniae* was cultured overnight in DMEM at 37˚C with agitation. After centrifugation ($8,000 \times g$, 30 min at 4˚C), the supernatant and bacterial pellets were collected. The bacterial pellets were lysed by sonication. The supernatant and the bacterial lysate were heated at 95˚C for 5 min. Caco-2 cells or RAW264.7 macrophages were incubated overnight at 37˚C with the bacterial supernatants, or lysate. The supernatant was collected, and Axl and Gas6 were detected using human Axl (Abcam, cat# 99976), human Gas6 (Invitrogen, cat# BMS2291), mouse Axl (Abcam, cat# ab100669), or mouse Gas6 (Abcam, cat# ab278087) ELISA kits.

## Statistical analysis

The data are presented as mean ± standard deviation (SD). The means of multiple groups were compared by analysis of variance, followed by Tukey's tests, using the JSTAT statistical software (version 8.2). Cumulative survival rates were analyzed using the Kaplan–Meier method, and differences in survival between subgroups were assessed using the log-rank test in SPSS version 22 for Windows (SPSS Inc., Chicago, IL, USA). The correlation coefficients (r) and their significance (p) were calculated between the variables using JSTAT statistical software (version 8.2). The animals were randomly assigned to various groups. For repeated immunostaining experiments, the conditions were randomized to account for potential ordering effects. Statistical significance was set at $p < 0.05$.

## Supporting information

**S1 Fig. Elderly mice infection model using *K. pneumoniae* ATCC43816 pmCherry. (A)** H&E staining of cecal mucosa of 15- or 57-week-old mice before initiation of the antibiotic treatment. This image has five independent replicates. Scale bar = 50 μm. **(B)** Mucosal thickness was measured by ImageJ analysis software. Each dot represents the value from an individual mouse (n = 5 per group). Data are presented as the mean ± SD. NS: not significant. **(C)** *K. pneumoniae* ATCC43816 were electroporated with the pmCherry plasmid. This image has seven independent replicates. Scale bar = 5 μm. **(D and F)** H&E staining of cecum (D) and liver (F) tissue from *K. pneumoniae* ATCC43816 pmCherry-infected mice aged 15 or 57 weeks. Desquamation of cecal epithelium is denoted by the black dotted line, and edema of the cecal submucosa is indicated by the red dotted line. Each image has six independent replicates. Scale bar = 50 μm. **(E)** Desquamation and edema length of cecal mucosa was measured by the ImageJ analysis software, and the percentage of desquamation or edema area were calculated. Each dot represents the value from an individual mouse (n = 6 per group). Data are presented as the mean ± SD. *$p < 0.05$. *p* values were calculated by the Student's *t* test. **(G)** Damaged area of liver per total area was measured and percentage calculated by ImageJ analysis software. Each dot represents the value from an individual mouse (n = 6 per group). Data are presented as the mean ± SD. *$p < 0.05$. *p* values were calculated by the Student's *t* test.
(TIF)

**S2 Fig. Caco-2 cells are injured by *K. pneumoniae* infection in the absence of RAW264.7 macrophages. (A)** H&E staining of *K. pneumoniae*-infected Caco-2 cells grown on the insert in the presence or absence of RAW264.7 macrophages. The cytoplasmic rupture of Caco-2 cells is indicated by the black dotted line. Each image has six independent replicates. Scale bar = 50 μm. **(B)** Desquamation length of Caco-2 cells grown on the insert was measured by the ImageJ analysis software and the percentage of desquamation area were calculated. Each dot represents six independent replicates (n = 6 per group). Data are presented as the mean ± SD. *$p < 0.05$. *p* values were calculated by the Student's *t* test.
(TIF)

**S3 Fig. The effect of the presence of macrophages on the expression of ZO-1 and occludin in Caco-2 cells without *K. pneumoniae* infection. (A and C)** Caco-2 cells grown on the insert in the presence or absence of RAW264.7 macrophages without *K. pneumoniae* infection were immunostained with an anti-E-cadherin antibody, and an anti-ZO-1 antibody (A) or an anti-occludin antibody (C). Each image has five independent replicates. Scale bar = 50 μm. **(B and D)** Fluorescence intensity per pixel of anti-ZO-1 antibody (B) or an anti-occludin antibody (D) were measured by the ImageJ analysis software. Each dot represents five independent replicates (n = 5 per group). Data are presented as the mean ± SD. NS: not significant.
(TIF)

**S4 Fig. The effect of Gas6/Axl signals on the expression of Axl, Gas6, ZO-1, and occludin in Caco-2 cells in the presence of BMDMs. (A)** *K. pneumoniae* infection model using a Transwell insert co-culture system based on Caco-2 cells and BMDMs were constructed. Prior to *K. pneumoniae* infection, Caco-2 cells were treated for 3 h with Axl inhibitor (R428; 20 nM) or 1 μg of anti-Gas6 antibody. Western blotting was performed to detect the expression of Axl, Gas6, ZO-1, and occludin in Caco-2 cells infected with *K. pneumoniae* in the presence of an Axl inhibitor (R428) or an anti-Gas6 antibody. Each western blotting image represents three independent replicates. **(B)** Western blotting signal intensity was analyzed by ImageJ software. Each dot represents three independent replicates (n = 3 per group). Data are presented as the mean ± SD. $*p < 0.05$, $**p < 0.01$. $p$ values were calculated by one way analysis of variance.
(TIF)

**S5 Fig. The effect of Gas6/Axl signaling on the expression of ZO-1 and occludin in Caco-2 cells without *K. pneumoniae* infection. (A)** Axl inhibitor (R428; 20 nM) or 1 μg of anti-Gas6 antibody were added to Caco-2 cells grown on the insert in the presence of RAW264.7 macrophages. Cells were immunostained with an anti-ZO-1 antibody and an anti-occludin antibody. Each image has five independent replicates. Scale bar = 50 μm. **(B)** Fluorescence intensity per pixel of anti-ZO-1 antibody or an anti-occludin antibody were measured by ImageJ analysis software. Each dot represents five independent replicates (n = 5 per group). Data are presented as the mean ± SD. NS: not significant.
(TIF)

**S6 Fig. Fluorescence intensity of immunostaining Gas6 and Axl in the cecum and liver of 15-weeks or 57-weeks-old mice infected with *K. pneumoniae*. (A and B)** Fluorescence intensity per pixel of anti-Gas6 or anti-Axl antibody of cecum (A) and liver (B) were measured by ImageJ analysis software. Each dot represents the value from an individual mouse (n = 6 per group). Data are presented as the mean ± SD. NS: not significant. $** p < 0.01$. $p$ values were calculated by the Student's $t$ test.
(TIF)

**S7 Fig. Western blotting signal intensity of Axl, Gas6, ZO-1, and occludin in Caco-2 cells treated with human Gas6 recombinant protein.** (A and B) Western blotting signal intensity (Fig 6B and 6C) was analyzed by ImageJ software. Each dot represents three independent replicates (n = 3 per group). Data are presented as the mean ± SD. $**p < 0.01$. $p$ values were calculated by one way analysis of variance.
(TIF)

**S8 Fig. The effect of Gas6 recombinant protein on the expression of ZO-1 and occludin in Caco-2 cells without *K. pneumoniae* infection. (A)** Gas6 recombinant protein (1 μg) was added to Caco-2 cells grown on the Transwell insert for 3 h. Cells were immunostained with an anti-ZO-1 antibody and an anti-occludin antibody. Each image has five independent

replicates. Scale bar = 50 μm. **(B)** Fluorescence intensity per pixel of anti-ZO-1 antibody, an anti-occludin antibody, and anti-E-cadherin antibody were measured by the ImageJ analysis software. Each dot represents five independent replicates (n = 5 per group). Data are presented as the mean ± SD. NS: not significant. $^*p < 0.05$. $p$ values were calculated by the Student's $t$ test.
(TIF)

**S9 Fig. Desquamation area of the cecal mucosal layer in 57-week-old mice infected with *K. pneumoniae* is reduced by administering Gas6 recombinant protein. (A)** H&E staining of cecum tissues from *K. pneumoniae* ATCC43816 pmCherry-infected mice. Desquamation of epithelial cells is indicated by the black dotted line. Each image has five independent replicates. Scale bar = 50 μm. **(B)** The area of desquamation was measured by ImageJ software. Each dot represents the value from an individual mouse (n = 5 per group). Data are presented as the mean ± SD. $^{**}p < 0.01$. $p$ values were calculated by one way analysis of variance. **(C and D)** Sections of cecal mucosa were from mice aged 15-week, 57-week, or Gas6 administered 57-week-old mice at 2 days after infection with *K. pneumoniae* ATCC43816 pmCherry and immunostained with an anti-F4/80 antibody. Each image has five or six independent replicates. Scale bar = 50 μm. Fluorescence intensity was analyzed by ImageJ software (D). Each dot represents the value from an individual mouse (n = 5 or 6 per group). Data are presented as the mean ± SD. $^{**}p < 0.01$. $p$ values were calculated by one way analysis of variance.
(TIF)

**S10 Fig. Westernblotting band images for Figs 4A, 5C, 6B, 6C, 7D, 7H, and S4A.**
(PDF)

**S1 Table. Source data for Figs 1B, 1C, 1E, 1G, 2B, 2C, 2F, 2H, 3A, 3B, 3C, 3E, 4B, 4D, 4F, 4G, 5D, 5E, 6E, 6G, 7B, 7C, 7E, 7G, 7I, S1B, S1E, S1G, S2B, S3B, S3D, S4B, S5B, S6A, S6B, S7A, S7B, S8B, S9B and S9D.**
(XLSX)

## Acknowledgments

We are grateful to the Support Center for Medical Research and Education at Tokai University School of Medicine for their technical assistance.

## Author Contributions

**Conceptualization:** Hitoshi Tsugawa.

**Data curation:** Hitoshi Tsugawa.

**Formal analysis:** Hitoshi Tsugawa, Takuto Ohki, Juntaro Matsuzaki.

**Funding acquisition:** Hitoshi Tsugawa, Juntaro Matsuzaki.

**Investigation:** Hitoshi Tsugawa, Takuto Ohki, Shogo Tsubaki, Rika Tanaka.

**Methodology:** Hitoshi Tsugawa, Takuto Ohki.

**Project administration:** Hitoshi Tsugawa.

**Supervision:** Hitoshi Tsugawa, Katsuto Hozumi.

**Writing – original draft:** Hitoshi Tsugawa.

**Writing – review & editing:** Takuto Ohki, Juntaro Matsuzaki, Hidekazu Suzuki, Katsuto Hozumi.

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
