## [Decision Letter · Decision Letter 0]

10 Feb 2023

Dear Dr Tsugawa,

Thank you very much for submitting your manuscript "Gas6 ameliorates intestinal mucosal immunosenescence to prevent the translocation of a gut pathobiont, Klebsiella pneumoniae, to the liver" for consideration at PLOS Pathogens. As with all papers reviewed by the journal, your manuscript was reviewed by members of the editorial board and by several independent reviewers. In light of the reviews (below this email), we would like to invite the resubmission of a significantly-revised version that takes into account the reviewers' comments.

If you wish to submit a revised manuscript, please provide a point by point response to each of the reviewers comments. Regarding the Major Issues raised by reviewer 1: Point 1, validation with a non-hypervirulent strain would be useful but not critical, this weakness can be specifically addressed in the text. Points 2,3 and 5 will require the inclusion of additional experimental data. Point 4 will require additional analysis of the data presented.

We cannot make any decision about publication until we have seen the revised manuscript and your response to the reviewers' comments. Your revised manuscript is also likely to be sent to reviewers for further evaluation.

Sincerely,

Matthew C Wolfgang

Academic Editor

PLOS Pathogens

David Skurnik

Section Editor

PLOS Pathogens

Kasturi Haldar

Editor-in-Chief

PLOS Pathogens

orcid.org/0000-0001-5065-158X

Michael Malim

Editor-in-Chief

PLOS Pathogens

orcid.org/0000-0002-7699-2064

If you wish to submit a revised manuscript, please provide a point by point response to each of the reviewers comments. Regarding the major issues raised by reviewer 1: Point 1, validation with a non-hypervirulent strain would be useful but not critical, this weakness can be specifically addressed in the text. Points 2,3 and 5 will require the inclusion of additional experimental data. Point 4 will require additional analysis of the data presented.

Reviewer's Responses to Questions

**Part I - Summary**

Reviewer #1: In this manuscript, Tsugawa and colleagues describe a multidisciplinary study of the role of Gas6 in the translocation of Klebsiella pneumoniae (Kp) from the gut to the periphery, specifically the liver. The authors use an aged mouse system to model the immunosenescence hypothesized to contribute to the susceptibility of elderly patients to Kp infection. The data are presented clearly and logically, and the authors deserve credit for the creative use of their aged mouse system. That said, there are several areas where the study and manuscript can be improved.

Reviewer #2: In the manuscript „ Gas6 ameliorates intestinal mucosal immunosenescence to prevent the translocation of a gut pathobiont, Klebsiella pneumoniae, to the liver" Tsugawa and colleagues combine infection experiments in antibiotic treated young and aged mice and an cell-culture based assay to study the impact of the protein Gas6 on the gut pathogen K. pneumoniae. They demonstrate that Gas6, which is released by intestinal macrophages, can recognize K. pneumoniae and thereby inhibits bacterial translocation through tight-junction barrier enhancement. As the amount of secreting macrophages decreases with age and administration of recombinant protein reverted the phenotype, the authors conclude that reduced Gas6 levels increase the susceptibility against K. pneumoniae in elderly mice. I think the findings are interesting and relevant to experts in the field. I think the manuscript is well written and generally convincing, although I do have some questions that I outline below, specifically regarding the performed animal experiments.

**Part II – Major Issues: Key Experiments Required for Acceptance**

Reviewer #1: 1. The use of strain ATCC43816 makes sense in that it is the most well-characterized Kp lab strain; however, this is a hypervirulent strain. Hypervirulent strains garner attention due to their ability to infect younger, healthy individuals. Although elderly individuals are also infected by hypervirulent strains, it would benefit this study to validate the authors' findings with a non-hypervirulent strain. These strains are more prevalent in healthcare settings, especially those with an elderly population.

2. Similarly, the in vitro data is generated entirely with the RAW264.7 and Caco-2 cell lines. Given that these cell lines are immortalized, this study would benefit from validation with primary cells, either mouse or human.

3. The data shown in Fig. 7C demonstrate that treatment with recombinant Gas6 reduced bacterial load in the cecum. The authors do not explore this finding in any detail. Does Gas6 have direct bactericidal effects in vivo? Or does Gas6 stimulate macrophage migration to the intestinal mucosa and reverse the effects seen in Fig 5E? This is important to understand, as it is unclear if the data shown in 7B are due to the effects of Gas6 on gut tight junction formation, as is posited throughout the manuscript, or simply due to the reduction in bacterial load.

4. There are several areas where images lack quantification (such as what is shown in Fig. 2D, 5C, and/or S3B) and experimental information. These are S1B, S2, Fig 1D-E, 3D, 4A, 5A-B, and 6B-C. For all images, please indicate the number of replicates these images represent.

5. Since the authors are activating their RAW264.7 cells with LPS, a no Kp control should be present in all relevant experiments, such as Figs. 2D-E, 4B-C, and 6D-E.

Reviewer #2: (No Response)

**Part III – Minor Issues: Editorial and Data Presentation Modifications**

Reviewer #1: 1. There are a few areas where the language can be refined:

a. Line 77 - Kp is not classically a gut pathogen. It is a gut colonizer with pathogenic potential. An example of a gut pathogen is C. diff.

b. Line 87 - Kp doesn't "target" the elderly. It is a more prevalent cause of infection in the elderly.

2. Please use log-scale y-axes in Figs. 1C, 2B, 3A, and 7C.

3. I would encourage the authors to discuss the data shown in Figs. 3B-C. Do the authors think that Kp LPS is the stimulating factor or is it some other factor?

4. Please indicate the oral inoculation method. Oral gavage, feeding, or other?

5. Please indicate the transwell pore size.

6. Please indicate if any blinding was performed prior to image analysis and how images were processed and selected for analysis.

Reviewer #2: Line 105 ff.: Which mouse line was used that carries a commensal K. pneumoniae strain? I am missing some information of the strain used to infect the mice? Which sequence type? Which capsule type? Is the strain hypervirulent, multi drug resistant or both? What are the reasons for choosing this strain? Is this strain relevant in the human population, especially with regard to infection of the elderly?

Line 118ff.: If the phenotype is really exclusively driven by the number of intestinal macrophages, the authors should exclude other potential influencing factors like age dependent differences in the microbiota, the mucosal architecture and other immune-related factors that has been described for K. pneumoniae-host immune system-interplay (Sequeira et al., 2020 DOI: 10.1038/s41564-019-0640-1, Constantinides et al., 2019 DOI: 10.1126/science.aax6624).

I would suggest to generate a more holistic data set of the used animals. How does the microbiome composition of the mice look before initiation of antibiotic treatment? Did the microbiome composition also change within the lifespan of the animals? More specifically, is the alpha diversity already reduced in elderly mice? If I understand correct, mice receive a heavy antibiotics cocktail over the whole course of experiment, which in turn would lead to a massive reduction of the present microbes in the gut and might affect the abundance and composition of relevant immune cell subsets. Which other bacteria, except of the commensal K. pneumoniae strain are actually still present? Is the status of being germfree validated? If the purpose is to completely eradicate the gut microbes, why not use germfree animals instead?

Why did the author chose a mouse model harboring an commensal K. pneumoniae strain in the gut? Is the immune system not already primed for K. pneumoniae then?

Figure 1+7: How many mice have been used? It looks like a single experiment has been performed with only 5 mice, which is in my point of view a very limited samples set. It would be worth repeating it at least once. It would be also interesting to see how different CFU counts of K. pneumoniae are in the lumen. If the number of microbial cells is already stronger reduced in younger mice, it would also mean that the general colonization resistance is general higher in younger mice meaning tissue infiltration is also less pronounced.

Also, in Figure 1, if mice were sacrificed on day 2 after infection, how could the authors monitor the survival over 16 days? (I don’t understand the cartoon in A and the resulting graph in B.

Have the authors checked the initial status of the mucosal architecture in the mice of different age before initiation of the antibiotics course? If the mucosal thickness is already different it would be also another explanation for the observed differences regarding tissue infiltration.

Line 114ff what about luminal vs tissue burden of K. pneumoniae? The data panels in figure 1 and 7 are relatively empty. I would generate some more data on the differences in the immune system of young and elderly mice.

How broadly applicable is this phenotype? Klebsiella pneumoniae strains are highly diverse (also stated by the authors). Does the exogenous application of Gas6 also prevents tissue infiltration of other hypervirulent K. pneumoniae strains? I would suggest to at least repeat the in vitro assay with another relevant human strain of K. pneumoniae.

Discussion

Line 250 the word symbiont implicates that the bacteria are beneficial for the host, better use pathobiont.

Line 284 the authors claim potential use of exogenous Gas6 to treat infection in the elderly community. But all the mouse models used by the authors rely on heavy antibiotics treatment, which in my point of view does not reflect the homeostatic conditions we find in the healthy community. Is this model also working in antibiotic naïve mice?

PLOS authors have the option to publish the peer review history of their article (what does this mean?). If published, this will include your full peer review and any attached files.

Reviewer #1: **Yes: **Jay Vornhagen

Reviewer #2: No
---

## [Decision Letter · Decision Letter 1]

29 Apr 2023

Dear Dr Tsugawa,

We are pleased to inform you that your manuscript 'Gas6 ameliorates intestinal mucosal immunosenescence to prevent the translocation of a gut pathobiont, Klebsiella pneumoniae, to the liver' has been provisionally accepted for publication in PLOS Pathogens.

Best regards,

Matthew C Wolfgang

Academic Editor

PLOS Pathogens

David Skurnik

Section Editor

PLOS Pathogens

Kasturi Haldar

Editor-in-Chief

PLOS Pathogens

orcid.org/0000-0001-5065-158X

Michael Malim

Editor-in-Chief

PLOS Pathogens

orcid.org/0000-0002-7699-2064

Reviewer Comments (if any, and for reference):

Reviewer's Responses to Questions

**Part I - Summary**

Reviewer #1: The authors have addressed my concerns.

Reviewer #2: (No Response)

Reviewer #3: The authors identiefied role of macrophages to combat opportunistic

Infection by by gut comensal bacteria such as K.pneumoniae! Identified

possible mechanism of high chances of K.neumoniae infection in aegeing individuals!

**Part II – Major Issues: Key Experiments Required for Acceptance**

Reviewer #1: (No Response)

Reviewer #2: The authors have adressed all concerns

Reviewer #3: None

**Part III – Minor Issues: Editorial and Data Presentation Modifications**

Reviewer #1: (No Response)

Reviewer #2: The authors have adressed all minor points

Reviewer #3: Not applicable

PLOS authors have the option to publish the peer review history of their article (what does this mean?). If published, this will include your full peer review and any attached files.

Reviewer #1: **Yes: **Jay Vornhagen

Reviewer #2: No

Reviewer #3: **Yes: **Prof .Dr Dwij Raj Bhatta, PhD

---

## [Editor Report · Acceptance letter]

10 May 2023

Dear Dr Tsugawa,

We are delighted to inform you that your manuscript, "Gas6 ameliorates intestinal mucosal immunosenescence to prevent the translocation of a gut pathobiont, *Klebsiella pneumoniae*, to the liver," has been formally accepted for publication in PLOS Pathogens.

Best regards,

Kasturi Haldar

Editor-in-Chief

PLOS Pathogens

orcid.org/0000-0001-5065-158X

Michael Malim

Editor-in-Chief

PLOS Pathogens

orcid.org/0000-0002-7699-2064